# Bipartite genome and structural organization of the parvovirus Acheta domesticus segmented densovirus

Judit J. Pénzes [1,2,4] ✉, Hanh T. Pham[1,5], Paul Chipman [2], Emmanuel W. Smith[3,6], Robert McKenna[2] ✉ & Peter Tijssen[1] ✉

Parvoviruses (family *Parvoviridae*) are currently defined by a linear mono-partite ssDNA genome, T = 1 icosahedral capsids, and distinct structural (VP) and non-structural (NS) protein expression cassettes within their genome. We report the discovery of a parvovirus with a bipartite genome, Acheta domes-ticus segmented densovirus (AdSDV), isolated from house crickets (*Acheta domesticus*), in which it is pathogenic. We found that the AdSDV harbors its NS and VP cassettes on two separate genome segments. Its vp segment acquired a phospholipase A2-encoding gene, *vp*ORF3, via inter-subfamily recombination, coding for a non-structural protein. We showed that the AdSDV evolved a highly complex transcription profile in response to its multipartite replication strategy compared to its monopartite ancestors. Our structural and molecular examinations revealed that the AdSDV packages one genome segment per particle. The cryo-EM structures of two empty- and one full-capsid population (3.3, 3.1 and 2.3 Å resolution) reveal a genome packaging mechanism, which involves an elongated C-terminal tail of the VP, "pinning" the ssDNA genome to the capsid interior at the twofold symmetry axis. This mechanism funda-mentally differs from the capsid-DNA interactions previously seen in parvo-viruses. This study provides new insights on the mechanism behind ssDNA genome segmentation and on the plasticity of parvovirus biology.

Parvoviruses (PVs) are small, non-enveloped icosahedral viruses, which infect vertebrate and invertebrate animals[1]. Densoviruses (DVs) are autonomously replicating, invertebrate-infecting members of the *Parvoviridae* family, classified into two subfamilies[2]. Members of the subfamily *Densovirinae* infect a wide array of terrestrial and aquatic invertebrates, in which they are pathogenic (reviewed by Penzes et al.[3]). The PVs of subfamily *Hamaparvovirinae* infect either vertebrates or invertebrates, with hamaparvoviral DVs classified into three genera. Members of the genera *Penstylhamaparvovirus* and

*Hepanhamaparvovirus* infect penaeid shrimps, and are pathogenic[3,4]. Members of the genus *Brevihamaparvovirus* infect exclusively mos-quitoes and are closely related to penstylhamaparvoviruses, sug-gested by their genome organization, protein homology and transcription strategy[5–9].

Members of the *Parvoviridae* have linear, monopartite ssDNA genomes of 3.6–6.2 kb[10], flanked by partially double-stranded, hairpin-like DNA secondary structures, which can form inverted terminal repeats (ITRs)[10,11]. The termini are essential for replication and genome

[1]Centre Armand-Frappier Santé Biotechnologie, Institut national de la recherche scientifique, Laval, QC H7V 1B7, Canada. [2]The McKnight Brain Institute, University of Florida, Gainesville, FL 32610, USA. [3]Institute of Molecular Biophysics, Florida State University, Tallahassee, FL 32306, USA. [4]Present address: Institute for Quantitative Biomedicine, Rutgers, the Sate University of New Jersey, Piscataway, NJ 08854, USA. [5]Present address: HTG Molecular Diag-nostics, 3430 E Global Loop, Tucson, AZ 85706, USA. [6]Present address: JEOL USA Inc., Peabody, MA 01960, USA. ✉e-mail: judycash08@gmail.com; rmckenna@ufl.edu; tijssenpc@gmail.com

packaging[10,11]. The parvovirus genome includes two expression cassettes, one of which encodes a varied number of non-structural proteins (NS). At least one of these proteins, conventionally designated NS1, contains a superfamily 3 (SF3) helicase domain, which includes the only highly conserved protein sequence motifs throughout the entire family[2,10]. The other expression cassette, designated *cap*, encodes one to four structural proteins (VPs). These are usually N-terminal extensions of one another, sharing an overlapping C-terminal segment, and are assembled into the capsid[10,12]. In case of subfamilies *Parvovirinae* and *Densovirinae*, the unique N-terminal extension of minor capsid protein 1 (VP1u), the largest of the VPs, typically encodes a highly conserved phospholipase A2 (PLA2) domain, essential for endosomal egress[13,14]. Following receptor-mediated endocytosis, in order to reach the nucleus to replicate, PVs have been shown to traffic through the endo-lysosomal system of the host cell, exposing the viral particle to increasing acidity from pHs 7.4–4.0[14–19]. Some parvoviruses, including all members of the *Hamaparvovirinae*, lack the PLA2 activity and use an alternate endosomal egress mechanism[19].

To date, there have been more than one hundred parvoviral capsid structures determined at or near atomic resolution, the majority belonging to the *Parvovirinae*, as opposed to only five derived from DVs. Three of these capsid structures belong to members of the *Densovirinae*, including Galleria mellonella densovirus (GmDV) at 3.7 Å resolution (PDB ID: 1DNV)[20] of genus *Protoambidensovirus*, Acheta domesticus densovirus (AdDV) at 3.5 Å resolution (PDB ID: 4MGU) of genus *Scindoambidensovirus*[21] and Bombyx mori densovirus (BmDV) at 3.1 Å resolution (PDB ID: 3POS) of genus *Iteradensovirus*[22]. The capsid structure of Penaeus stylirostris densovirus (PstDV) at 2.5 Å resolution (PDB ID: 3N7X) of genus *Penstylhamaparvovirus*[23] is the only high-resolution structure of the *Hamaparvovirinae* to date. The fifth structure belongs to the divergent Penaeus monodon metallodensovirus (PmMDV) at 3 Å resolution (PDB ID: 6WH3), lacking a subfamily affiliation[19]. All PV structures possess T = 1 icosahedral symmetry, comprised by 60 VP subunits. Each subunit displays an eight-stranded (βB to βI) jelly roll fold[24], in which variable loops link the β-strands together to compose the variable capsid surface morphology[12]. With the exemption of PmMDV, the interior jelly roll BIDG sheet is complemented by an additional N-terminal β-strand, canonically designated βA[12,19]. The fivefold symmetry axis of the PV capsid characteristically includes a pore-like opening that continues in a channel, a portal proposed to aid genome packaging and uncoating, as well as for PLA2 domain externalization[25–27].

The common house cricket (*Acheta domesticus*) is a host to 2 DVs of subfamily *Densovirinae*. AdDV is highly pathogenic, causing mass mortality at cricket rearing facilities worldwide[28,29]. The AdDV harbors an ambisense genome, which includes a "split" VP gene, a *Scindoambidensovirus* characteristic. Consequently, its VP1 is encoded by a spliced transcript, in which the PLA2-containing VP1u is expressed by a separate ORF (*cap1*) upstream from *cap2*, the latter ultimately giving rise to VPs 2–4 via leaky scanning[28]. Acheta domesticus mini ambidensovirus (AdMDV), of genus *Miniambidensovirus*, also harbors an ambisense expression strategy, but only includes one *cap*, encoding a PLA2-encompassing VP1[30].

Reported is the discovery, complete genome sequence, transcription strategy and near-atomic 3D structure of a previously undescribed DV infecting the common house cricket, designated Acheta domesticus segmented densovirus (AdSDV). Thus far, AdSDV is the first PV to harbor a bipartite genome, a result of recombination between subfamilies *Densovirinae* and *Hamaparvovirinae*. We show that each segment is packaged into a separate viral particle, to maintain particle size and integrity of its *Brevihamaparvovirus* ancestors. The AdSDV has a complex transcription strategy, which is distinct from other members of its genus *Brevihamaparvovirus*, a potential adaptation to the multipartite replication strategy. Furthermore, AdSDV relies

on a unique DNA-packaging model, which involves both the threefold and twofold axes and results in increased thermostability of the full virions, by reinforcing the twofold axis via direct stacking interactions between the interior wall and the ssDNA genome. This study of AdSDV provides new perspectives on parvoviral genome and transcription evolution as well as on capsid architecture.

## Results

### Virus discovery and pathogenesis

In February of 2013, common house crickets at an insect rearing facility in Ontario, Canada, exhibited clinical signs consistent with a viral infection i.e., erratic, uncontrolled movement, followed by paralysis and death. Icosahedral viral particles, ~220 Å in diameter, could be visualized in homogenized fat bodies of affected specimens by negative staining electron microscopy (Fig. 1a). The presence of known DVs or CRESS DNA viruses of similar size and morphology were excluded by PCR testing. Following CsCl density gradient purification, the particles were introduced via oral administration of contaminated food to healthy house cricket nymphs. These developed identical signs in 14 to 20 days as the original outbreak, whereas direct injection of the purified particles into the abdominal fat body accelerated the progression of the disease by ~7 days. These experiments were repeated involving two more commercially reared cricket species i.e., *Gryllus bimaculatus* and *Grylloides sigillatus*, but neither displayed signs of infection or harbored viral particles in homogenized fat bodies. Isolated DNA could not be amplified by rolling circle amplification, which suggested a linear genome. Consequently, the extracted DNA was blunt-ended, cloned and sequenced.

### Complete genome characterization and phylogeny inference of a previously undescribed densovirus with a segmented genome

Sequencing identified two cloned populations, both ~3.3 kb in size. To verify whether both were present in the extracted viral DNA, native, blunted DNA was subjected to restriction endonuclease (RE) digestion by enzymes with recognition sites only in one (HindIII, SpeI) or in both (XbaI) obtained sequences (Fig. 1b). The resulting restriction profiles supported the presence of a heterogenous, bipartite DNA isolate. Both segments (segment I and II) were flanked by T-shaped hairpin-like secondary structures of 208- and 220 nt, respectively, which did not form ITRs but were identical at the corresponding termini of both segments, implying a common genome origin (Fig. 1c). Segment I (3316 nt in length) harbored three complete and two partial ORFs (Table S1, Fig. 1d). The derived amino acid (aa) of segment I identified two proteins of 796 and 379 aa in length, respectively, which displayed significant similarity with the NS1 and NS2 of various mosquito-infecting brevihamaparvoviruses according to a BLASTP (basic local alignment search tool protein)[31] search (NS1: 41% identity at 69% coverage, Haemagogus equinus DV; NS2: 41% identity at 97% coverage, Aedes albopictus DV 2), hence the segment I was designated the *ns* segment. The longer ORF, now referred to as NS1, harbored an SF3 helicase domain. The 26-aa-long nsORF3 did not harbor any detectable homology to any known protein. Segment II (3332 nt) included three ORFs (Table S1, Fig. 1e) and was designated the *vp* segment, as *vp*ORF1 was predicted to encode a 378-aa-long protein, a homolog of a *Brevihamaparvovirus* VP from Aedes albopictus DV 2 (40% identity at 83% coverage). The putative 104-aa-long product of the small central ORF, *vp*ORF2, displayed no significant homology with any GenBank entry. The 347-aa-long putative product of *vp*ORF3 was identified as a homolog of the VP1u-encoding *cap1* of AdDV (45% identity at 67% coverage). Consequently, ORF3 also harbors a PLA2 domain, similarly to the *Scindoambidensovirus* VP1u. Segment 2. Due to the bipartite genome, this DV was named Acheta domesticus segmented densovirus (AdSDV).

Based on the phylogeny inference of the family-wide conserved SF3 helicase domain, AdSDV clustered to genus *Brevihamaparvovirus*

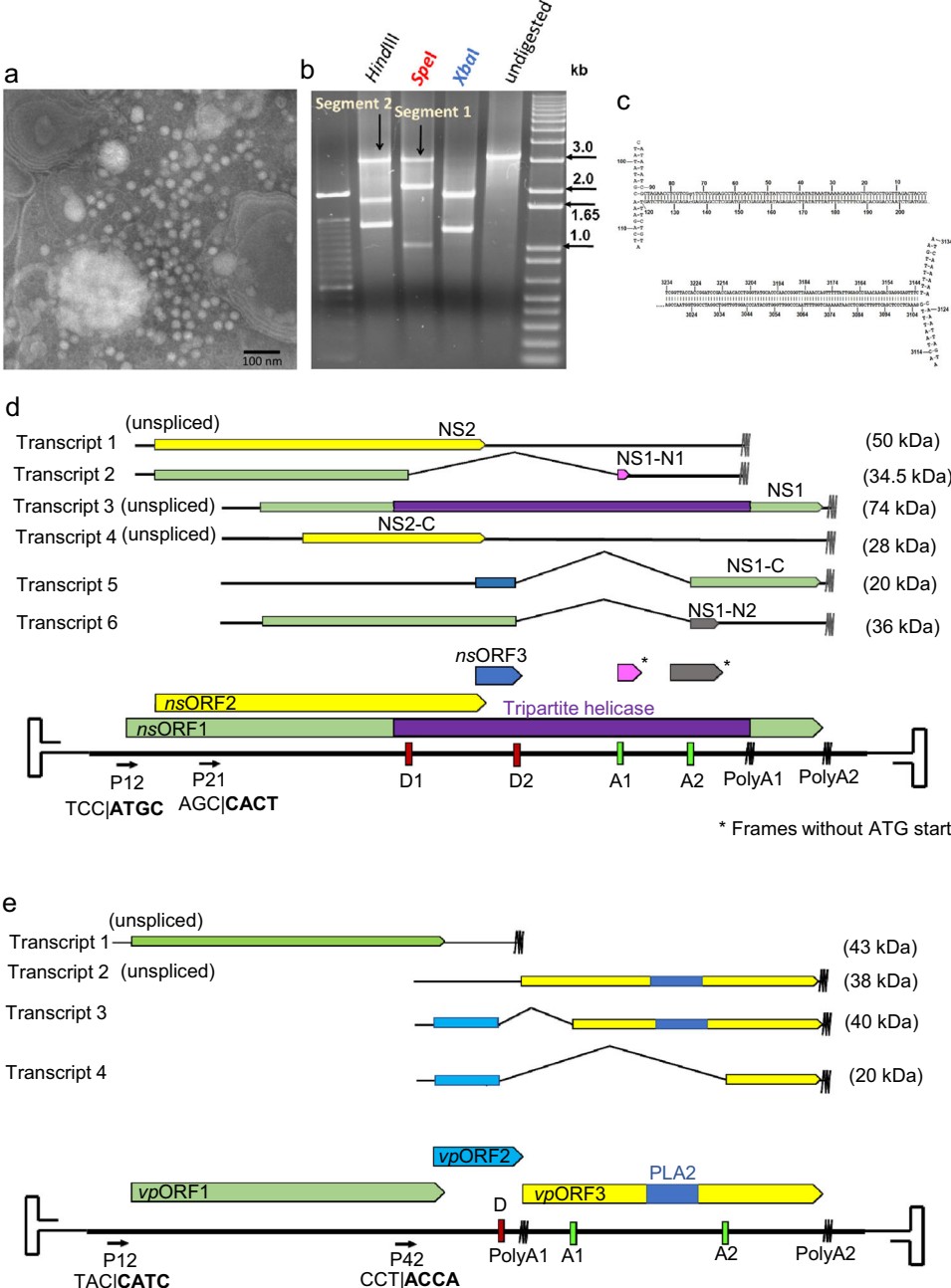

**Fig. 1 | Discovery, genome organization and transcription strategy of Acheta domesticus segmented densovirus (AdSDV). a** Icosahedral virus particles visualized in the homogenized fat bodies of infected common house crickets (*Acheta domesticus*) by negative staining transmission electron microscopy. This is one representative micrograph of ten taken at the time, each displaying the particles in similar numbers. **b** Ethidium bromide-stained agarose gel image displaying the results of digestion by restriction endonucleases cutting only the NS-, VP segment, or both. This is one representative ethidium bromide-stained agarose gel of three such experiments, displaying the same results. **c** DNA secondary structure predictions of the T-shaped genome termini, flanking both the NS and VP segment. **d** Genome organization and transcription strategy of the NS segment. Open reading frames (ORFs) are marked by the colored arrows and boxes, mRNA is stylized by the black lines. Small open reading frames, which function as exons, but lack a canonical ATG start codon, are marked by an asterisk (*). The wavy lines mark polyadenylation signals. Promoters are labeled as "P," donor- and acceptor sites as "D" and "A" respectively. Below the promoters the sequence of the transcription start site is shown, with the actual mRNA 5' end presented in bold behind the vertical line. The estimated molecular weights of the protein products to be expressed by each transcript are shown to the right. **e** Genome organization and transcription strategy of the VP segment, using the same display and labeling scheme as in **d**.

of subfamily *Hamaparvovirinae*, becoming the first non-mosquito infecting member and the first to harbor a PLA2 domain (Fig. 2).

## Transcription strategy

To characterize the transcriptome of the bipartite genome, total RNA was isolated from infected house crickets, three days following direct fat body inoculation, then reverse-transcribed and subjected to

specific amplification by PCR. Both segments harbored two promoters and two polyadenylation signals, respectively (Fig. 1d, e). The upstream *ns* segment promoter at map unit 12 (P12) yielded two transcripts and both are polyadenylated at the proximal polyadenylation signal (positioned at 2551 nt, tail is added at 2612 nt). Transcript 1 is capable of expressing the complete NS2 in its entire length and did not undergo splicing. The spliced transcript 2 utilized the upstream donor site (D1,

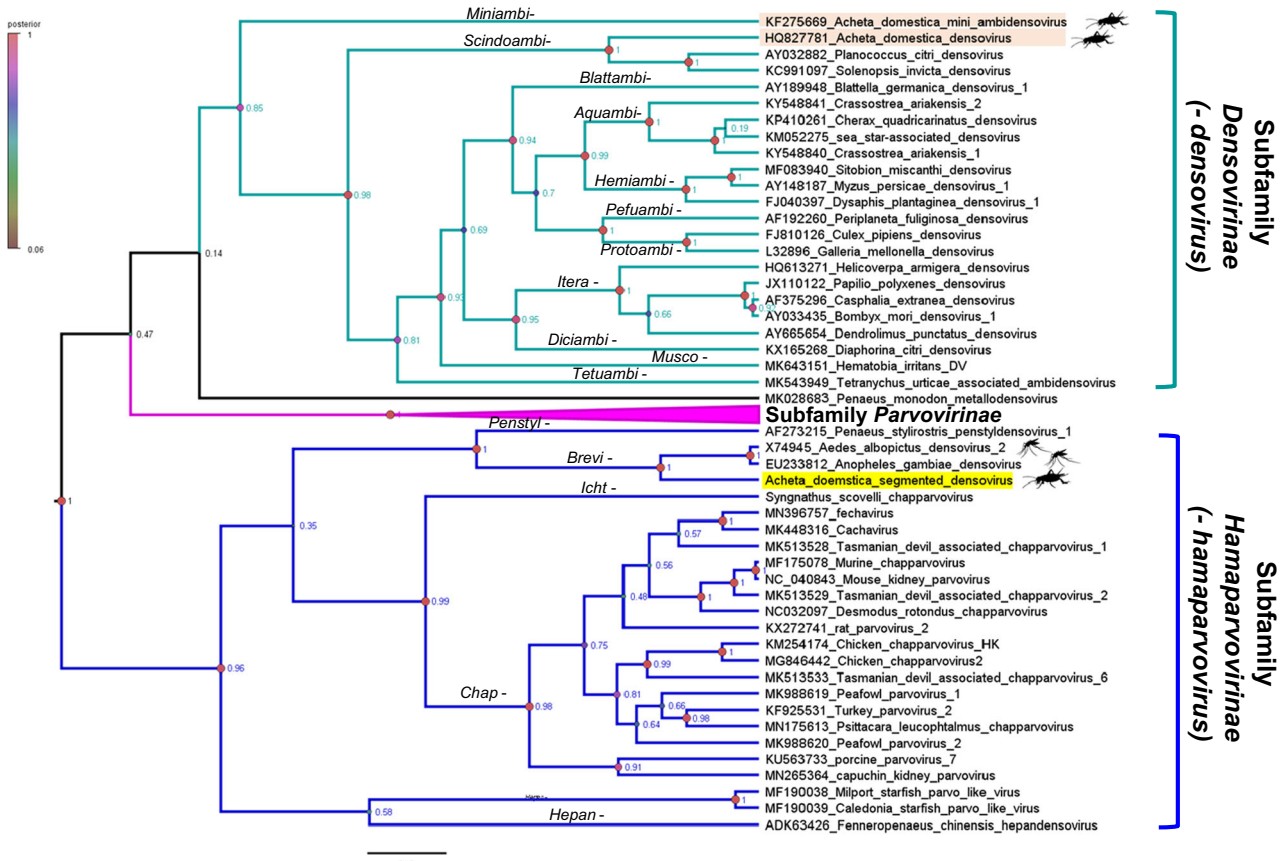

**Fig. 2 | Bayesian phylogeny inference based on a 162-aa-long region of the NS1 protein, corresponding with the superfamily 3 helicase domain, the only protein sequence conserved throughout the *Parvoviridae* family.** Each sequence represents one species, while the *Parvovirinae* subfamily is collapsed for visualization. Genera names are shown on the branches and the posterior probability values to evaluate the reliability of the topology are shown as node labels. A color coding of the node shapes is also displayed, according to the posterior probability values. House cricket infecting densoviruses are highlighted in apricot and labeled with the silhouette of the animal. Acheta domesticus segmented densovirus clusters within the mosquito-infecting genus *Brevihamaparvovirus* and is highlighted in yellow.

ATC | CA), linking it with the proximal acceptor site (A1, TTATCAA | AA), removing NS-intron 1 (Table S2, Fig. S3). This results in an ORF, NS1-N1, which can express the N-terminal region of NS1, terminating the frame directly upstream of the SF3 domain, by receiving a 7-aa-long tail from a small ORF without an ATG start codon. There are four NS transcripts transcribed from the downstream promoter, P21, terminating at the distal polyA signal (positioned at 2865 nt, polyA tail is added at 2880 nt). The unspliced transcript 3 is capable of expressing an N-terminally truncated NS1 ORF, translated from the second ATG of the original frame, located in a strong Kozak context (CCGCC**ATG**G). With a predicted molecular weight of 74 kDa, this is the only NS segment-derived protein, which includes the complete SF3 helicase domain. Either from this transcript, or from a putative transcript 4, an N-terminally truncated NS2 product (NS2-C), could also be expressed, using the second ATG codon of the NS2 frame, located in a weaker Kozak context (ACAC**ATG**A). The spliced transcript 5 utilized a distinct set of donor and acceptor sites from those of transcript 2, namely D2 (GCA | AG) and A2 (GGGAGCA | GAG) (Table S2, Fig. S3). The removal of NS-intron 2 puts in frame a small auxiliary ORF, ORF3, with the C-terminal portion of the NS1 ORF, potentially encoding the 20-kDa-sized NS1-C. As for transcript 4, the existence of transcript 6 as a separate mRNA population is debatable. The removal of the same intron results in the union of the NS1 ORF with a 25-aa-long tail from another ATG-lacking small ORF, giving rise to NS1-N2. Based on the transcription profile, the AdSDV *ns* segment has a potential coding capacity for six putative NS proteins.

The *vp* segment also encoded two promoters (P12, P42) and two polyadenylation signals, resulting in the separation of the *Brevihamaparvovirus*-like ORF1 and the *Scindoambidensovirus*-like ORF3, together with auxiliary ORF2, into two separate expression units. The expression of *vp*ORF1, from unspliced transcript 1 is under the control of the upstream P12 and is polyadenylated at the PolyA1 site, directly upstream from the *vp*ORF3 ATG start codon (at position 1844 nt, polyA tail added at pos. 1852 nt). Transcript 2, the only unspliced transcript of the downstream P42, is potentially capable of expressing the PLA2-encoding ORF3. The spliced transcripts 3 and 4 shared the same donor site (GAA | GA) but used separate acceptor sites, A1 (TATTATA | AAC) and A2 (CAAAAAA | GAC), respectively (Fig. S3). While transcript 3 unites *vp*ORF2 with an almost complete *vp*ORF3, the removal of the longer intron from transcript 4 only preserves the C-terminal portion of this ORF, removing the PLA2-encoding region.

**Structural and nonstructural proteins from the VP segment**
Using the Bac-to-Bac baculovirus expression system, Sf9 cultures were transfected by three recombinant bacmid constructs, namely AdSDV-Bac-*vp*ORF1 and AdSDV-Bac-*vp*ORF3, with a polyhedrin promoter-linked *vp*ORF1 or *vp*ORF3, respectively, as well as AdSDV-*vp*-P42, containing the entire P42-promoter-linked expression unit in a poly-hedrin promoter knock-out construct. Virus-like particle (VLP) formation was only observed in the AdSDV-Bac-*vp*ORF1-transfected culture. The VLPs were purified utilizing a sucrose step gradient, with particle accumulation in the 20% fraction (Fig. 3a). Transfection, using

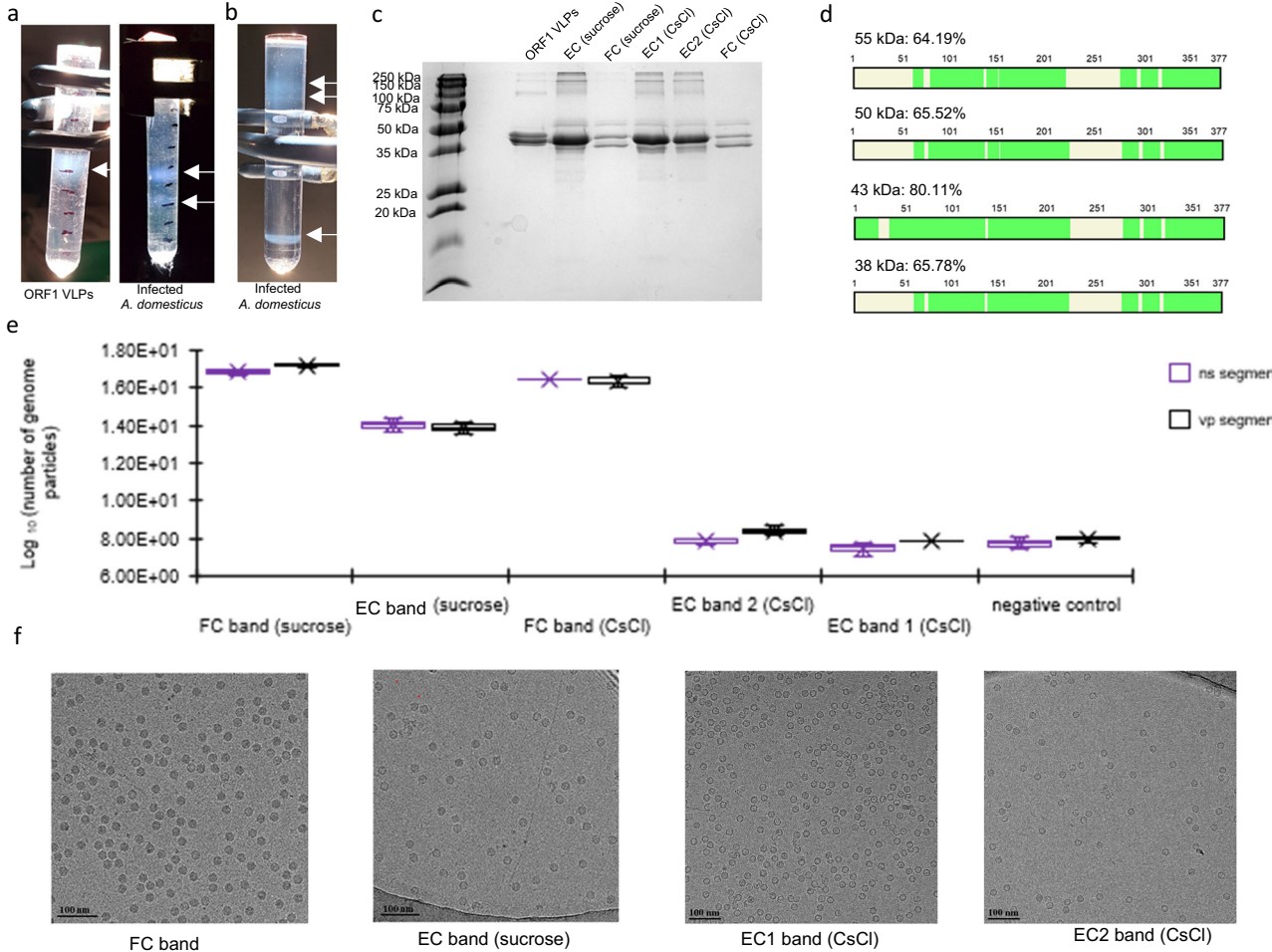

**Fig. 3 | Purification and protein analyses of the Acheta domesticus segmented densovirus (AdSDV) capsids. a** Sucrose step gradients visualized under fluorescent light, following ultracentrifugation. Fractions occupied by purified AdSDV virus-like particles (left) or capsids (right) are marked by the small white arrows. **b** Purification of AdSDV capsids directly from the deceased house crickets by a continuous CsCl gradient, which was also used to assess the buoyant density of each capsid population. The small arrows mark the AdSDV capsid fractions. **c** SDS-PAGE in 15% polyacrylamide of the obtained sucrose- and CsCl gradient-purified fractions. EC stands for empty capsids, FC marks genome-filled full capsids, ORF1 VLPs were expressed recombinantly from the vpORF1 structural protein gene. **d** Coverage map of the protein sequencing reads, obtained by Nano-liquid chromatography tandem mass spectrometry (Nano-LC/MS/MS) and previously excised from the polyacrylamide gel in panel C. Regions that were represented among the MS reads are colored green. **e** Box and whiskers plots showing the quantification of AdSDV viral DNA in each purified capsid fraction, using real-time PCR (qPCR). The obtained genome copy numbers are presented on a logarithmic scale of the basis of 10. Whiskers indicate the range of the data points, with the maximum and minimum values shown at the ends. Box walls indicate the lower- and upper quartile, respectively. The median is indicated by the central box line. The mean of each series is shown by an X. Source data are provided as a Source Data file for this panel. $N = 3$ quasi-independent experiments (virus populations from which DNA was extracted derive from three purification event of the same batch of AdSDV-infected common house crickets). **f** Electron micrographs of each AdSDV capsid fraction.

the AdSDV-Bac-*vp*ORF3 bacmid led to rapid cytopathic effect (CPE) within two days post-transfection, as opposed to five to seven days in case of AdSDV-Bac-*vp*-ORF1. Using DEAE-dextran, AdSDV-Bac-*vp*ORF3 was also transfected to live common house crickets, which all deceased within three days post-inoculation, as opposed to not developing any signs upon transfection with AdSDV-Bac-*vp*ORF1 in the 14-day-long observation period.

Capsids were purified from the lethargic AdSDV-infected house crickets. The subsequent sucrose gradient had two distinct particle bands; in the 20% fraction (empty capsids, EC), and in the 25% fraction (genome containing "full" capsids, FC) (Fig. 3a). When subjected to isopycnic centrifugation in a continuous CsCl gradient, the EC band separated into two bands, with a buoyant density of 1.132 g/cm³ (EC1) and 1.191 g/cm³ (EC2), respectively, while the single FC band had a buoyant density of 1.459 g/cm³ (Fig. 3b).

Analysis by SDS-PAGE revealed a variance in the incorporation ratio of protein bands at sizes of ~55, 50, 43, 40 and 38 kDa (Fig. 3c) in the capsid populations. The bands were excised and analyzed by Nano-

liquid chromatography tandem mass spectrometry (Nano-LC/MS/MS), and the protein sequences were searched against the NCBI non-redundant protein database as well as against the AdSDV genome, revealing that all bands comprised solely products of *vp*ORF1 (Table S4). The 43-kDa-sized SDS-PAGE band corresponded with the predicted weight of *vp*ORF1 and was the only one with coverage throughout the complete ORF (Fig. 3d). Consequently, this protein was designated VP1, the major component of the EC, EC1 and EC2 capsids and ~50% of the FC population. All the minor bands displayed the same coverage profile, being N-terminally truncated versions of VP1. We designated the 38-kDa-sized protein VP2, representing a minority fraction of the EC, EC1 and EC2 capsids, yet accounting for half of the VPs comprising the FC particles. VP2 was also the component of the 55 and 50 kDa minor bands, the size of which exceeds the coding capacity of the AdSDV genome.

The EC and FC fractions varied significantly in genome content, yet each contained a similar ratio of ns and vp segments (Fig. 3e). Cryoelectron microscopy (cryo-EM) micrographs revealed that the

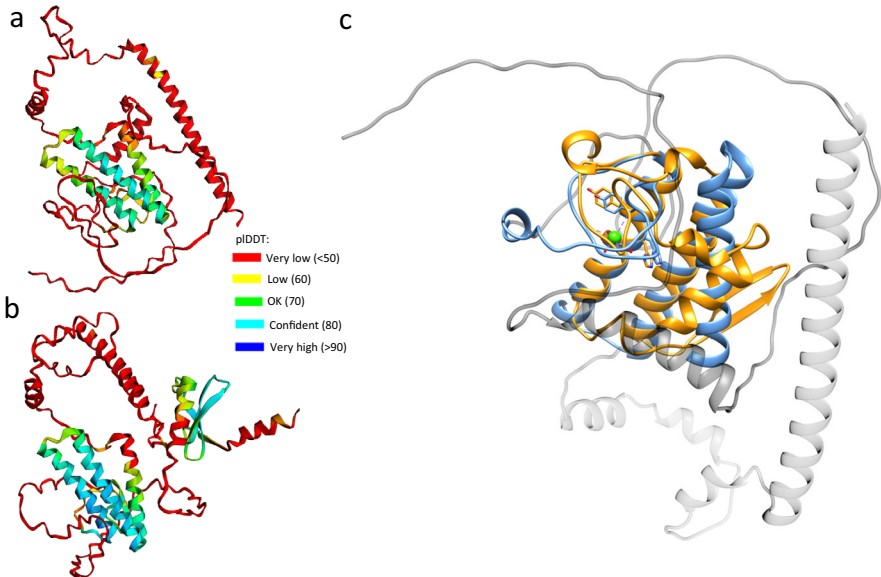

**Fig. 4 | Relation of Acheta domesticus segmented densovirus (AdSDV) vpORF3 and secretory Phospholipase A2 (sPLA2) enzymatic proteins.** Structural predictions of the Acheta domesticus segmented densovirus (AdSDV) vpORF3 protein (**a**), as well as of the protein product of vp-transcript 3, expressed by joining vpORF2 N-terminally with vpORF3 (**b**). Predictions were created by AlphaFold2 pipeline provided by ColabFold (https://www.biorxiv.org/content/10.1101/2021.08.15.456425v1). The ribbon diagrams in both panels are colored by the pLDDT values, indicating the reliability of the prediction as shown by the legend. **c** Shows the vpORF3 model phospholipase A2 (PLA2) domain (blue) superimposed with that of the Chinese cobra (*Naja atra*) venom PLA2 structure (PDB ID:1POA) (orange). The atoms and bonds are shown for the Tyr sidechain of the conserved YXGXG domain of the PLA2 calcium binding loop, as well as the sidechains of the conserved catalytic core (HD). The Ca²⁺ ion, present in the cobra venom PLA2 structure, is marked by a green sphere.

consistently high genome count of the FC population (in the range of >10¹⁵ genome particles of each segment) is indeed due to the presence of almost exclusively full, genome-packaging particles (Fig. 3f). This number was multiple logs lower in case of the EC capsids, which displayed an overwhelmingly large proportion of empty capsids. The CsCl-purified EC1 and EC2 bands were composed of exclusively empty particles (Fig. 3f). To further investigate the absence of the PLA2-including *vp*ORF3 products, a colorimetric PLA2 assay was performed involving each capsid population and it was shown that none of these displayed PLA2 activity, in concordance with the vpORF3 absence suggested by the Nano-LC/MS/MS (Fig. S5). Homology modeling of vpORF3 (Fig. 4a) and its spliced derivate (Fig. 4b) by AlphaFold2[32] revealed that the protein harbors a highly conserved PLA2 catalytic core, displaying structural similarity to PLA2 proteins, such as the 119-aa-long venom protein of the Chinese cobra (*Naja atra*) (Fig. 4c). The homology models of *vp*ORF3, as well as its spliced derivatives, did not display structural resemblance to canonical viral capsid proteins.

## Structural studies

The AdSDV capsid populations and VLPs were subjected to data collection by cryo-EM followed by single- particle image reconstruction (Table S6), obtaining capsid structures for the ORF1-VLPs at 3.3 Å (PDB ID: 8EU7), for the EC (PDB ID: 8ERK), EC1 (PDB ID: 8EU6) and EC2 (PDB ID: 8EU5) capsids at 2.5 Å, 3 Å and 3.1 Å, respectively, as well as for the FC capsids at 2.3 Å (PDB ID: 8ER8) (Fig. 5a). Each population possessed the expected T = 1 icosahedral symmetry and a particle size of 20 to 25 nm. The AdSDV capsid displayed an overall "smooth" capsid surface with the only protruding area surrounding the fivefold symmetry axis. The interior of the EC1 and EC2 was devoid of density, apart from a small amount of "dust" in the proximity of the threefold axis. The same location was occupied by a larger amount of disordered density in the *vp*ORF1-VLPs. The interior of the FC was filled with density attributed to the ssDNA genome. The slight amount of genome-like density of EC capsid interior confirmed the genome quantification and EM results, namely that this population is heterogenous and is composed of EC1, EC2 and FC capsids (detailed by Fig. S7).

The *vp*ORF1 sequence could be modeled into the FC density map from Thr47 to Leu377, the final C-terminal residue (Fig. 5b). The AdSDV VP subunit displayed the canonical eight-stranded jelly roll fold, complemented by an N-terminal βA, located on an elongated N-terminal region. The arrangement and approximate size of each loop from the βA to βB through to βH to βI is most similar to that of PstDV, supported by a DALI Z-score of 16.8[12]. The EC, EC1, EC2 and the ORF1-VLPs monomer lacked two N-terminally ordered residues as well as the unusually elongated C-terminal tail of the FC population, making Gln366 the last ordered residue (Fig. 5c). When superimposing the monomer VP model of all capsid populations, the only region exhibiting significant conformational difference was the DE loop, which comprises the fivefold channel and wall (Fig. 5c). The VP capsid models built into the EC1 and EC2 density, respectively, were essentially identical (Fig. 6a).

Although AdSDV displays a distinct morphology within the *Parvoviridae*, the overall topology of its exterior surface resembles that of the PstDV capsid (Fig. 7a), superimposable with a root-mean square deviation (RMSD) of 2.8 Å (Fig. 7b). The AdSDV surface morphology is significantly different from that of AdDV, with only their jelly roll cores superimposable (Fig. 7b). The AdSDV capsid harbors the smallest *Parvoviridae* interior volume (Fig. 8a), which is very apparent, given its capsid is able to fit within the GmDV capsid interior (Fig. 8b).

## Multimer interactions

The AdSDV fivefold channel displayed two distinct conformation states i.e., either an "empty" for the EC fractions and the *vp*ORF1-VLPs, or "filled" as observed in the FC population (Fig. 9a). In the filled state, the two additional ordered residues connect the density column to the FC shell, inferring the density to be part of the disordered 46-aa-long N-terminal region. While the channel of the *vp*ORF1-VLPs is covered by a hydrophobic plug, small, diffuse density occupies the channel in the EC1 and 2 populations. In a low, 10 Å resolution map, the EC2 channel is revealed to be significantly narrower than its EC1 counterpart (Fig. S6B). To accommodate the density column, the peak of the FC DE loop moves tangentially away from the channel, which is occupied by large

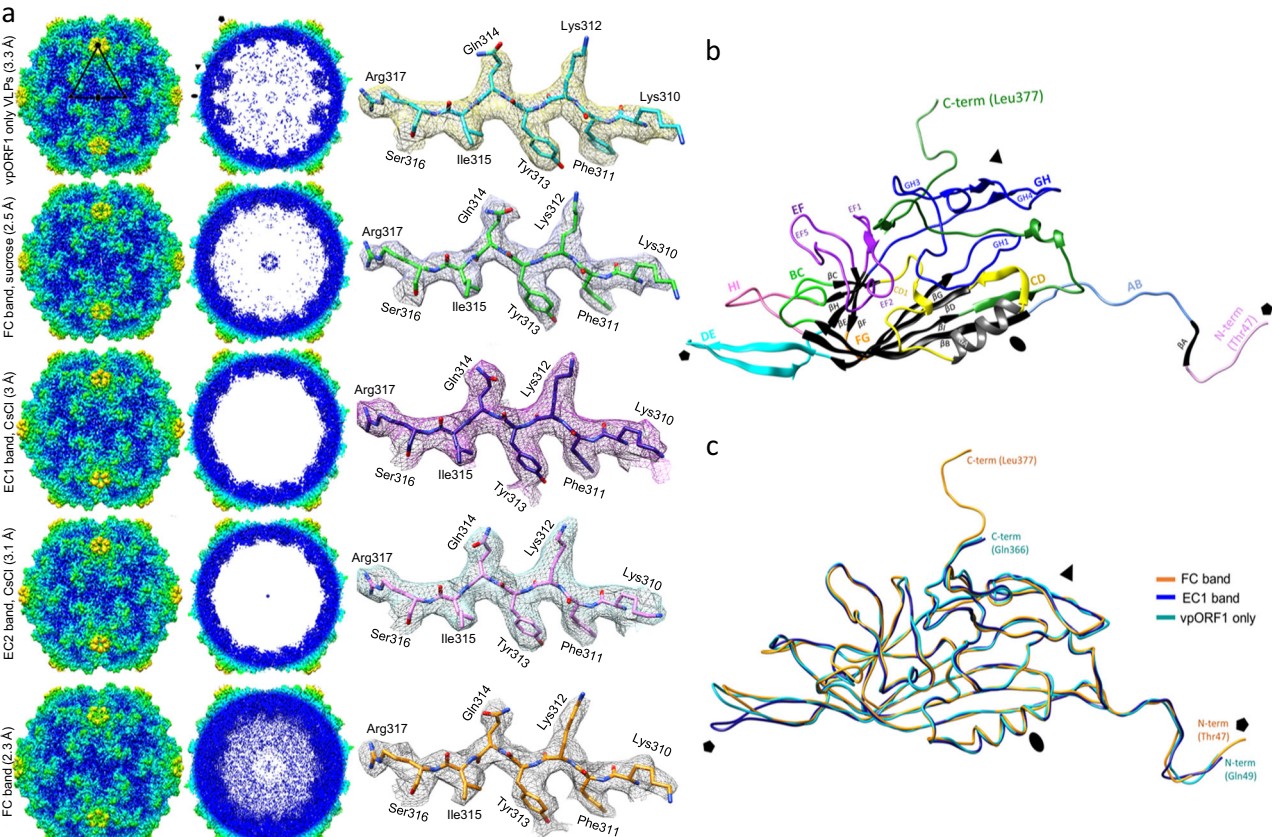

**Fig. 5 | Cryo-EM single particle structural studies of the Acheta domesticus segmented densovirus (AdSDV) capsids and vpORF1 virus-like particles (VLPs).** **a** Surface (left) and cross-section (right) views of the obtained AdSDV capsid and vpORF1 VLPs electron density maps, rendered at σ=1, with the example density and atomic model shown next to each map at σ = 4. The maps are colored radially, orientated in the I1 icosahedral convention (twofold axis in z plane). The five- three- and twofold symmetry axes are marked by a pentagon, triangle, and ellipse, respectively. The full, genome containing capsid fraction is labeled FC, while EC abbreviates the various empty capsid fractions pulled from the sucrose and isopycnic CsCl gradients. **b** Ribbon diagram showing the atomic model of a single subunit from the AdSDV full capsid (FC). Structurally conserved elements are shown in black and gray, the loops connecting these are highlighted in their respective colors. Symmetry axes are labeled the same as in panel A, to indicate orientation. **c** Superimposition of the subunit atomic model ribbon diagrams, obtained from the genome-filled infectious capsid population (FC), from the empty capsid fraction with the highest-buoyancy (EC1) and the vpORF1, recombinantly expressed VLPs.

aromatic and hydrophobic sidechains, such as Tyr159 and Ile152 (Fig. 9b). In contrast, both EC populations displayed DE loops pointing inwards, narrowing the pore from 16.0 Å (FC) to 10.1 Å, with Tyr159 and Ile152 retracted underneath the DE loop peak. The *vp*ORF1-VLPs possess a similar conformation to that of the FC capsids (Fig. 9b).

The AdSDV capsid harbors a second pore at its threefold axis, which is 10.8 Å wide and created by three interwoven β-strands, forming a β-annulus (Fig. 9c), lined by large, hydrophobic sidechains, covering a ring of three histidines (His232).

The long N-terminal segment βA of the AdSDV capsid subunits is in a domain-swapping conformation at the twofold symmetry axis, with its βA interacting with the βG of its twofold-neighboring subunit, comprising the interior surface of five-stranded β-sheets in a BIDGA order (Fig. 9d).

**Capsid-genome interactions**

The elongated C-terminal portion of the AdSDV-FC proteins enters the capsid interior from the threefold symmetry axis, connecting to the inner surface throughout the twofold interface. Each subunit interacts with the C-terminal tail of the fivefold neighbor to their twofold neighboring subunit (Fig. 10a). This interaction is absent from all the other capsid populations, resulting in the last ordered C-terminal residue to enter the interior directly below the threefold symmetry axis, where the free-hanging C-terminus manifests as various amounts

of disordered density (Fig. 10b, Fig. 5a). The ordered density of six nucleotides per each FC subunit occupies the interior twofold axis. Five of these are interlinked and interact directly with the capsid surface via π-stacking interactions (Fig. 10c). The first stack (stack 1) also incorporates the sixth free-standing purine nucleotide. The ordered ssDNA displays the sequence of purine-purine-pyrimidine-purine-purine, a GC-rich motif, which is especially abundant in the AdSDV termini as well as at various intervals throughout the entire sequence of both segments (85 times for the NS segment, 75 times for the VP segment). The genome density displayed an increasing amount of order when it is in close proximity to the interior surface, as the result of the C-terminal tails "grabbing" and "sticking" the genome to the inner twofold axis, by establishing π-stacks via the GC-rich penta-nuclear motifs (Fig. 10d).

Genome packaging also altered the thermostability of the AdSDV capsid, characterized by differential scanning fluorometry (DSF) (Fig. 9e, melt curve profiles shown by Fig. S8). At neutral pH the FC capsids already possessed a 2–4 °C higher unfolding temperature compared to the other populations. This increases to 5-6 °C at pH 6.0, simulating the environment of the early endosome, and to 6–8 °C at pH 5.5, modeling the environment of the late endosome. At the lysosomal pH of 4.0, the difference is still 3–4 °C. Regardless of genome content, each AdSDV capsid population displayed peak stability at pH 5.5, which declined significantly at the pH 4.0 of the mature lysosomes.

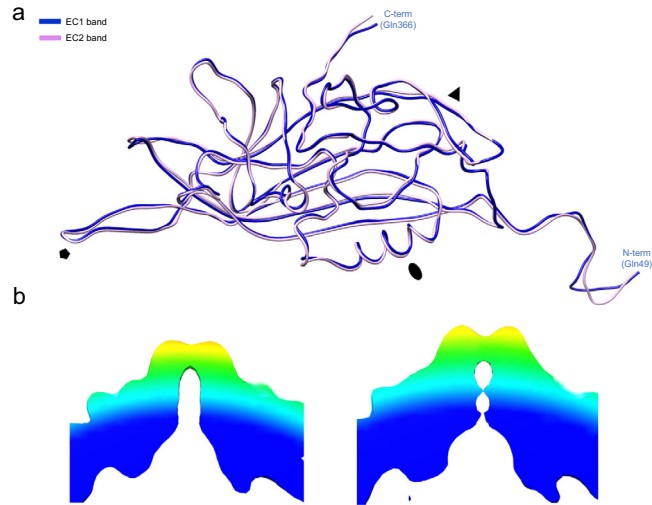

**Fig. 6 | Structural comparison of the two empty capsid bands (EC1 and EC2, respectively), pulled from the continuous CsCl gradient. a** Structure of the EC1 and EC2 monomers are identical, except for highly flexible, poorly modeled regions. **b** Low-resolution structural reconstruction of the EC1 (left) and EC2 (right), showing the cross-section of the fivefold channel, at the resolution of 8.4 Å and 9.3 Å, respectively ($\sigma = 1$). The narrowing of the EC2 fivefold channel could not be observed in the high resolution cryo-EM structures.

## Discussion

As currently defined, all *Parvoviridae* members possess a ssDNA monopartite genome, as one of the key characteristics of the family[10]. Despite of its bipartite genome, AdSDV displays all the other characteristics required for classification within the *Parvoviridae* (see Introduction). Its distinct NS and VP cassettes, however, are uniquely located on two separate genome segments.

Genome segmentation is rare among DNA viruses of animal hosts, unlike multipartite DNA viruses of fungi and plants, which package each DNA segment into a separate viral particle[33]. Experiments by Ojosnegros et al.[34] suggested for ssRNA+ viruses that genome segmentation allows maximizing genetic content while preserving capsid stability and energetically favorable genome density. Phylogeny inference showed that AdSDV is a member of the *Hamaparvovirinae* genus, *Brevihamaparvovirus*. The *Brevi-* and *Penstylhamaparvovirus* genera harbor the smallest genomes within the *Parvoviridae*, which are packaged into similarly small particles[23]. The AdSDV capsid interior volume is ~50% of that of GmDV, yet their buoyant density is similar (1.44 vs 1.45 g/cm³)[20]. As the united length of its two segments would only be slightly larger, yet would significantly increase particle density, it is reasonable to assume that AdSDV is also a multipartite-multicompartment virus. Another arthropod-infecting linear ssDNA virus family, the *Bidnaviridae*, also harbors a bipartite genome, with genes derived from four distinct viral lineages[35]. Bidnaviruses, however, despite of encoding PV-like capsid proteins, utilize a distinct replication strategy by encoding a DNA-dependent DNA polymerase

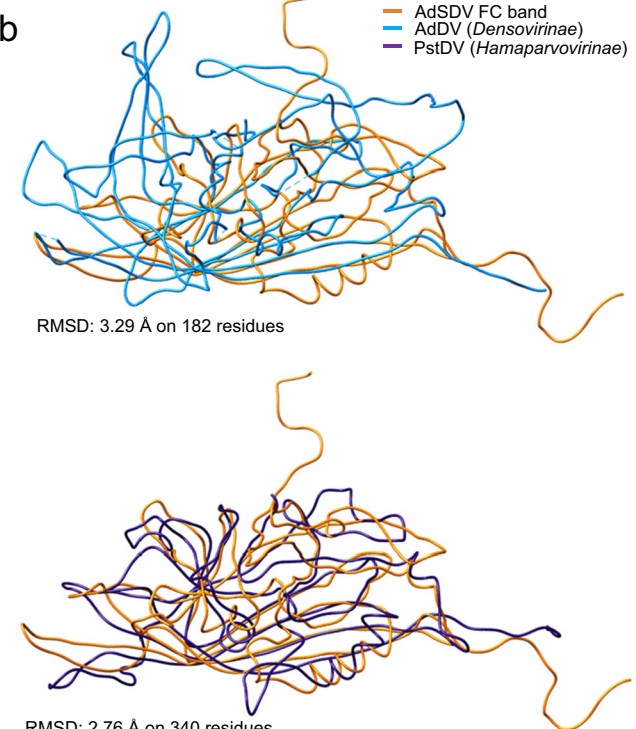

**Fig. 7 | Comparative analysis of the Acheta domesticus segmented densovirus (AdSDC) capsid structure with those of the *Parvoviridae* family. a** Capsid surface comparison of the AdSDV full capsid (FC) with those of other members of the *Parvoviridae* family i.e., invertebrate-infecting Penaeus stylirostris densovirus (PstDV), Penaeus monodon metallodensovirus (PmMDV), Acheta domesticus densovirus (AdDV), Bombyx mori densovirus (BmDV) and Galleria mellonella densovirus (GmDV) as well as vertebrate-infecting canine parvovirus (CPV) and adeno-associated virus 2 (AAV2). Each surface model map is orientated the same way as in panel A and is radially colored. **b** Ribbon diagram superimposition of the AdSDV FC subunit atomic model with those of another house cricket infecting densovirus from the *Densovirinae* subfamily (AdDV) as well as with PstDV, its closest relative for which the high-resolution capsid structure has been resolved, of subfamily *Hamaparvovirinae*.

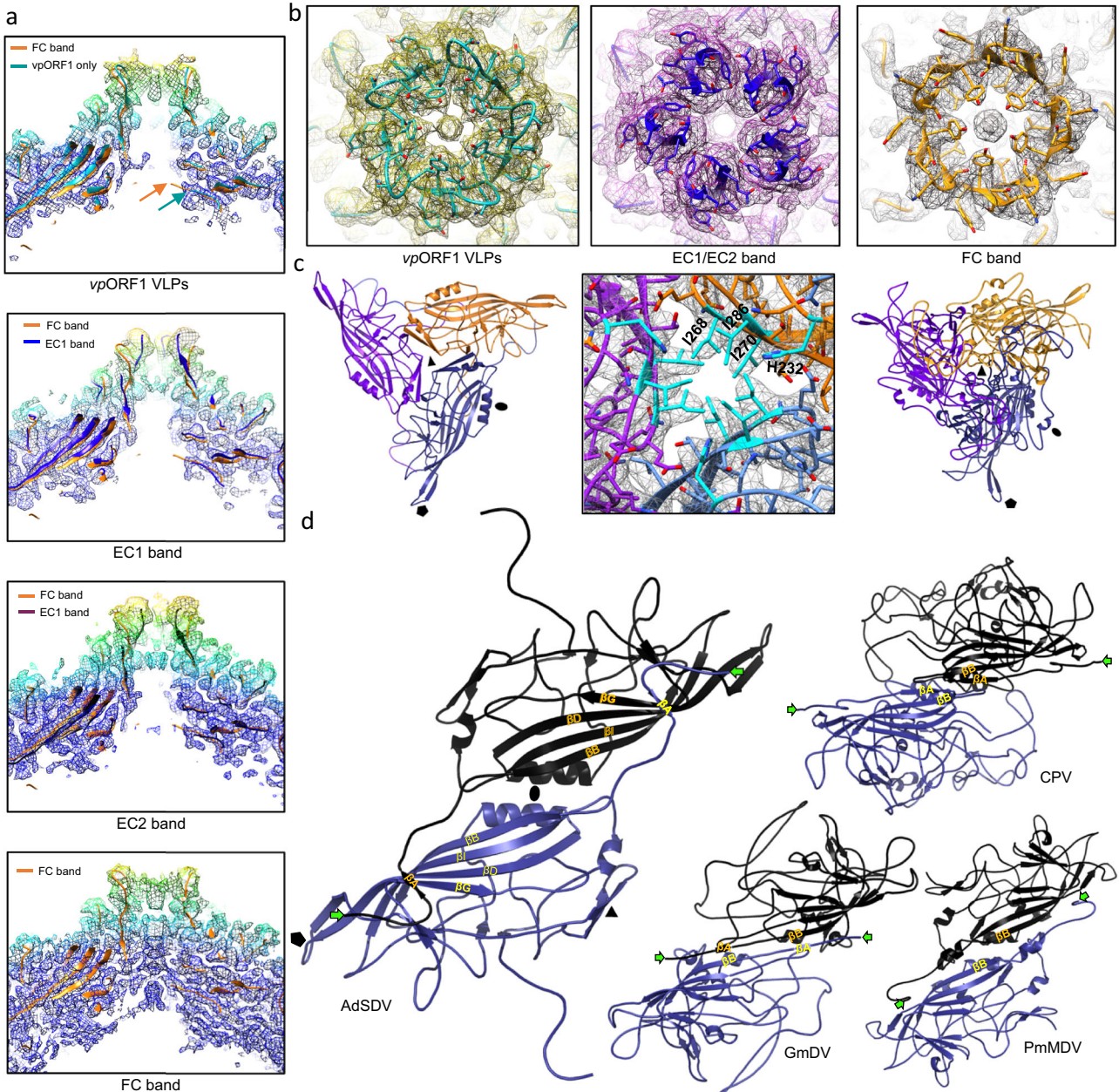

**Fig. 8 | Multimer interactions of the Acheta domesticus segmented densovirus (AdSDV) capsid. a** Cross-section view of the fivefold channel in case of the *vp*ORF1 virus-like particles (VLPs), both empty capsid (EC) populations and the full, genome-filled particles (FC). The electron density is radially colored from the map center and is shown as a mash at σ = 2. Each map is fitted with its corresponding atomic model as ribbon diagrams as well as with the ribbon diagram of the FC structure. The orange arrow indicates the FC N-terminus. **b** Top-down view of the opening of the fivefold channel, the fivefold pore, with the electron density rendered at σ=2. The atomic model fitted shows the ribbon diagram as well as the sidechains of the corresponding residues. Note the drastic conformation change of the five DE loops in opening up the channel from the closed conformation of the empty capsids (EC) vs. the open conformation of the genome-filled FC population. Note the difference in the hydrophobic plug covering the channel in case of the *vp*ORF1 VLPs, as opposed to the actual N-terminus externalization observed in the FC capsids. **c** Ribbon diagrams of the AdSDV FC trimer (left panel), displaying the opening of the β-annulus, typical of densoviruses. The middle panel shows the hydrophobic and positively-charged sidechains, highlighted in cyan, occupying the annulus, and the right panel shows an example of the threefold axis architecture of vertebrate-infecting parvoviruses of the *Parvovirinae*, represented by canine parvovirus. **d** Ribbon diagrams of the AdSDV dimer inner surface, interior in comparison with the three other types of dimer assembly strategy, described in the *Parvoviridae* thus far. Symmetry axes are labeled by a pentagon (fivefold axis), a triangle (threefold axis) and an ellipse (twofold axis). The N-terminus of each subunit is marked by the green arrow. Note the differences between the vertebrate-infecting parvoviruses, represented here by canine parvovirus (CPV) in comparison to the domain swapping conformation of the invertebrate-infecting members of the family, represented by Galleria mellonella densovirus (GmDV) and Penaeus monodon metallodensovirus (PmMDV).

gene in their genomes and initiate replication via protein priming[36]. The AdSDV genome is the first to display inter-subfamily recombination within the *Parvoviridae*, by incorporating the *Densovirinae*-originated *vp*ORF3 into its ancestral hamaparvoviral genome. Acquiring distant ORFs to increase viral fitness may lead to eventual genome

segmentation and a multipartite replication strategy in linear ssDNA viruses to maintain particle stability and optimal genome density.

To date, all parvoviral PLA2 domains are either located within structural protein-encoding genes or obtain the common C-terminal VP region via alternative splicing, so that they can be assembled into

## a

| | Inner radius (Å) | Inner surface area (nm²) | Inner volume (nm³) | Genome size (nt) | Genus |
|---|---|---|---|---|---|
| CPV | 92.9 | 1085 | 3360 | 5323 | *Protoparvovirus* |
| AAV2 | 89.9 | 1015 | 3040 | 4679 | *Dependoparvovirus* |
| GmDV | 98.7 | 1,223 | 4,023 | 6039 | *Protoambidensovirus* |
| AdDV | 91.8 | 1,056 | 3,228 | 5425 | *Scindoambidensovirus* |
| BmDV1 | 98.7 | 1224 | 4,028 | 5076 | *Iteradensovirus* |
| PstDV | 87.6 | 963 | 2,811 | 3914 | *Penstylhamaparvovirus* |
| AdSDV | 82.2 | 848 | 2322 | 3332 / 3316 | *Brevihamaparvovirus* |
| PmMDV | 86.5 | 933 | 2710 | 4371 | *Incertiparvovirinae* |

## b

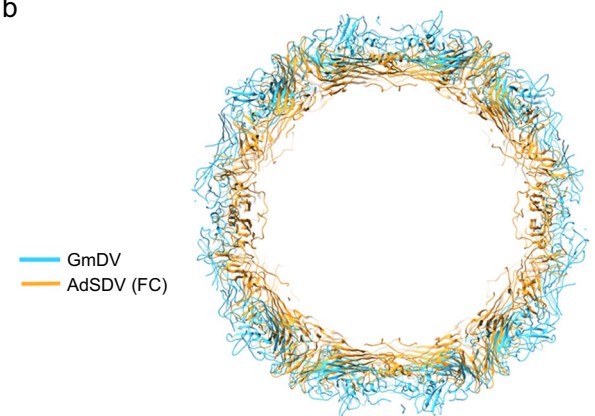

— GmDV
— AdSDV (FC)

**Fig. 9 | Acheta domesticus segmented densovirus (AdSDV) harbors the smallest parvoviral interior. a** Comparing the capsid and genome metrics of various members of the *Parvoviridae*, with those of (AdSDV). Subfamilies, including the *Parvovirinae*, *Densovirinae* and *Hamaparvovirinae*, respectively, are separated by the thick lines. Abbreviations: CPV canine parvovirus, AAV Adeno-associated virus 2, GmDV Galleria mellonella densovirus, AdDV Acheta domesticus densovirus, BmDV1 Bombyx mori densovirus 1, PstDV Penaeus stylirostris densovirus, PmMDV Penaeus stylirostris metallodensovirus. The inner volume of BmDV1 is probably overestimated since the residues of VP3 were partially traceable (372 out of 454 residues) in the icosahedrally averaged electron density (except for 42 N-terminal and 40 C-terminal residues). **b** Direct comparison of the AdSDV capsid size with that of the largest parvoviral capsid so far, for which the atomic structure is resolved, GmDV. Both full capsid models are shown as ribbon diagrams and the central cross-section view is shown here.

the capsid shell[12,28,37]. In both instances the PLA2 domain is located upstream of the major VP-encoding region. *vp*ORF3, of *Scindoambidensovirus* origin, is not only placed downstream of the major *vp*ORF1, but is also included in a separate transcription unit. Consequently, AdSDV might be a unique parvovirus to express a nonstructural PLA2. Secreted and cytosolic PLA2s exist in a wide range of organisms and are associated with cell lysis[38] and play a role in initiating apoptosis[39]. This may imply that *vp*ORF3, with a conserved catalytic core and Ca²⁺ binding loop, could potentially contribute to the cellular egress and spread of AdSDV, especially when the toxicity of the individually expressed VP-ORF3 is considered. This raises the question whether this domain in parvoviruses with a structural PLA2 also has a dual function.

Parvoviral transcription strategies vary significantly even within each subfamily, with vertebrate-infecting PVs and ambisense DVs relying heavily on alternative splicing[3,10]. Brevihamaparvoviruses, however, display a simpler transcription profile, relying exclusively on leaky scanning[40]. AdSDV, in contrast, harbors a complex transcription strategy, which employs alternative splicing. Ambisense members of the *Densovirinae* also possess a higher number of NS proteins, sometimes as many as four[3,41]. Ambisense and multipartite genomes both may overcome the limitations of the typical parvoviral temporal

monosense promoter expression order by potentially allowing the transcription machinery simultaneous access to the VP and NS expression cassettes. The large number of AdSDV NS proteins hence might be an adaptation to the multipartite replication strategy, which suggests extreme flexibility in parvoviral expression evolution, maximizing the coding capacity of the small genome.

The surface morphology and size of the AdSDV capsid, unlike its transcription profile, remained reminiscent of PstDV, a hamaparvovirus. However, the AdSDV capsid displays multimer interactions unlike other PVs. The N-termini of DV VPs are arranged in a domain-swapped conformation, where the βA interacts with the βB of the twofold-neighboring subunit and not with its own βB, as in case of the *Parvovirinae*[20–23]. In both instances, however, the capsid interior surface is composed of the five-stranded ABIDG sheets[12]. The AdSDV dimer represents yet a third type of interior architecture, which also relies on five-stranded β-sheets, but of BIDGA conformation instead.

The DV threefold axis is covered by the β-annulus, which creates an opening on the capsid surface of various size, similarly to the *T* = 3 ssRNA virus family, *Tombusviridae*[42] rather than the *Parvovirinae*, where this area is covered by spikes and protrusions[12]. This opening was previously suggested to be the location of DV genome packaging, in the GmDV capsid[20]. The AdSDV annulus is also lined by large hydrophobic sidechains as well as bulky, positively charged residues, as opposed to the abundant negative charge of the fivefold pore entrance, the canonical location of DNA entry and uncoating[12]. Moreover, the elongated 12-aa-long portion of the C-termini are also located here, in the absence of a packaged genome. This region is not involved in comprising the AdSDV shell, yet directly interacts with both the interior surface and the genome. These characteristics suggest a threefold axis-related genome-packaging model, instead of a fivefold-involving one.

Little is known about how the genome interacts with the PV interior. Protoparvoviruses harbor large portions of icosahedral-ordered ssDNA, arranged in π-stacks, while adeno-associated viruses and AdDV display only a couple of ordered nucleotides at the threefold axis[16,21,43–47]. Interactions between the interior and these nucleotides, however, are scarce and limited to potential hydrogen bonds. The AdSDV FC structure displayed the highest number of ordered nucleotides thus far, which directly interact with the interior capsid surface. This interior-genome interaction, which involves four subunits along with two separate regions of the ssDNA genome, provides additional stability to the otherwise weak interactions at the twofold axis. This mechanism may be responsible for the increased thermostability of the FC population. Regardless of genome content, the AdSDV capsids display an identical pH-linked thermostability profile to PLA2-including members of the *Parvovirinae*, which suggested a similar endo-lysosomal trafficking pathway, even in the absence of a capsid-bound PLA2 domain[16,25,27,48].

The electron density occupying the parvoviral fivefold channel has been consistently associated with N-terminal externalization, following either exposure to low endosomal pH or as the result of genome packaging[25,43,49]. The AdSDV is the first parvovirus, where the "density column", occupying the fivefold channel of the FC fraction, harbors a direct connection to the ordered N-terminus, confirming that N-terminal externalization is a conserved mechanism and independent from the presence of a functional PLA2.

The genome-packaging FC, in contrast with the EC populations, comprised a VP1-to-VP2 ratio of 1:1. It is possible that some of the externalized VP N-termini undergo proteolytic cleavage, which would explain the VP incorporation ratio shift. This mechanism, provided it exists, might have evolved to ensure that only matured virions reach the eventual replication site, as several virus families require cellular proprotein convertases to mature[50]. Unlike mature capsids of the FC population, the EC segregated into two populations of buoyant density. As the structure of the EC1 and EC2 fractions only differ in the

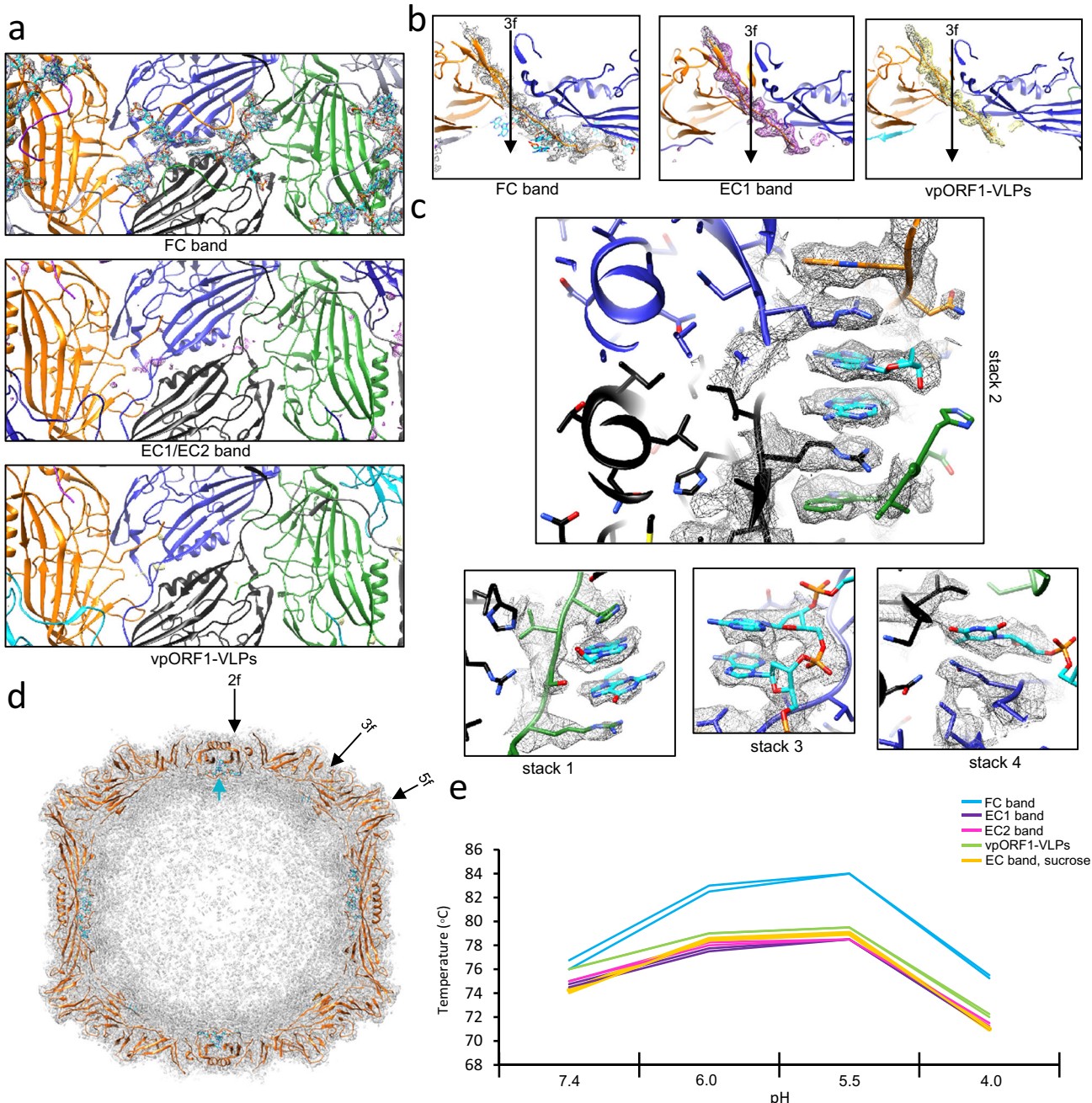

**Fig. 10 | Interactions between the Acheta domesticus densovirus (AdSDV) ssDNA genome and the capsid interior. a** Interior view of the AdSDV twofold axis, with the representation of electron density zoned to the six ordered nucleic acid bases (σ=3), only present in case of the genome-filled full capsids (FC). The ribbon diagram representation of the chain A atomic model is shown in blue and its twofold-neighboring subunit in black. The fivefold-neighboring subunit of chain A is presented in green, while the fivefold- neighbor of its twofold neighboring subunit in yellow. The atomic model of the nucleic acid is shown in cyan. **b** Side cross section view of the β-annulus, occupying the AdSDV threefold axis. The actual threefold axis is marked by an arrow. Electron density is shown zoned to the final C-terminally ordered residues, indicating that the ordered portion of the FC C-terminus stretches far and bends underneath the twofold symmetry axis. In the absence of a packaged genome, the ordered region ends directly underneath the threefold axis. **c** The AdSDV genome interacts with the interior surface via π-stacking interactions, underneath the twofold symmetry axis. Note how these stacks are the result of interactions between two ssDNA regions and four subunits, colored the same as in **a**. **d** Cros-section of the AdSDV FC map, with density shown as a mash at σ=1.5. The closer the interior genome density is located to the twofold symmetry axis, the more ordered it appears. The cyan arrow points to the location of the ordered nucleotides under the twofold symmetry axis. **e** Melting temperature profile of the AdSDV capsids and VLPs at the four pHs parvoviruses may encounter during endo-lysosomal trafficking, performed in duplicates. Note the difference in thermostability between the empty particles and VLPs (EC1, EC2, EC, *vp*ORF1-VLPs) vs. the genome-containing FC population. Results of *n* = 2 independent runs are shown. Source data are provided as a Source Data file for this panel.

conformation of the fivefold channel, these populations might differ in stages of particle maturation. Alternatively, the EC2 capsids might chelate ions, which could not be averaged icosahedrally, similarly to the GmDV capsid[20]. Although the *vp*ORF1-VLPs lack a packaged genome, their fivefold pore displays the open conformation of the FC particles. This suggests that the conformation changes leading up to opening the fivefold pore might happen even prior to packaging-induced N-terminus externalization.

Taken together, AdSDV provides the first evidence of a multi-partite replication strategy within the *Parvoviridae*, as the consequence of maintaining a *Brevihamaparvovirus*-like capsid size and morphology yet harnessing the fitness gain from a recombinant ORF from another subfamily. Adaptations imposed by the multipartite replication strategy of AdSDV may manifest in its expression profile, requiring a higher number of NS proteins, which could have led to the incorporation of alternative splicing. AdSDV also demonstrates a unique DNA-packaging mechanism, which might involve the threefold annulus for genome entry, after which the genome is attached to the interior twofold axis by the VP C-termini. When packaging is complete, the VP N-termini undergo externalization through a possibly already open fivefold channel, which subjects them to proteolytic cleavage at some point of their life cycle. These findings alter the perspectives of parvoviral traits previously deemed conservative, including a monopartite genome, a capsid-associated PLA2 domain, parvoviral capsid-DNA interactions and DV interior architecture.

## Methods

### Virus detection, infection, DNA isolation and cloning

Deceased common house crickets (-100 individuals of mixed gender) from the rearing facility were mechanically homogenized in 1x phosphate buffered saline (PBS) (137 mM NaCl, 2.7 mM KCl, 10 mM Na$_2$HPO$_4$, 1.8 mM KH$_2$PO$_4$, pH 7.4). The homogenate was cleared up by low-speed centrifugation and PBS-diluted supernatant was applied on a glow-discharged carbon-covered Cu grid (Electron Microscopy Sciences) and stained with 2% uranyl acetate. The dried grids were visualized at an acceleration voltage of 120 kV.

From this sample, virus particles were purified by cesium chloride (419.5 mg/mL) gradient ultracentrifugation to obtain viral DNA for cloning. By chloroform/butanol (1:1 volume) extraction, followed by low-speed centrifugation, a clear supernatant containing viral particles was obtained. Virus stock was concentrated from the supernatant by ultracentrifugation at 40,000 rpm in a type 60Ti rotor for 2 h at 4 °C. Pellets were resuspended in small volume of PBS and was again checked by negative staining EM to verify particle presence. This virus stock was applied to dried cricket feed in a 0.1 mg/mL protein concentration. Using a 1 mL insulin syringe and a delicate needle, -5 μl was injected into -10-mm-long house crickets, targeting the fat bodies inside the abdomen but avoiding the puncture of abdominal organs.

Viral DNA was extracted by the High Pure Viral Nucleic Acid Kit (Roche) and eluted in 40 μM (μL) of distilled water. The extracted DNA was subjected to amplification by Phi29 DNA polymerase (New England Biolabs). The isolated DNA was blunt-ended utilizing T4 DNA polymerase and Large Klenow fragment of DNA polymerase I (New England Biolabs) in the presence of 33 μM of each dNTP and cloned into the EcoRV restriction site of a pBluescript KS+ vector and sequenced by primer walking. The complete sequence of both segments was cloned, but without intact termini. To obtain the sequences of the termini, single-stranded adapters (5'−Phos−ATCCACAACAACTCTCCTCCTC−3') were linked to the AdSDV genome segments using T4 RNA ligase I (New England Biolabs). Using the reverse adapter primer paired with another primer, targeting the AdSDV genome in proximity of the termini, PCR amplification was performed. The 25 μL amplification reaction utilized Phusion® high-fidelity polymerase (New England Biolabs), supplemented by 10% dimethyl sulfoxide and 3 μL 2 mM EDTA. The obtained amplicons were blunt-cloned into an EcoRV-digested pBluscript KS+ plasmid and transformed into the Sure Escherichia coli strain (Stratagene), cultured at 30 °C.

To construct recombinant bacmid DNA, the VP regions to express were PCR-amplified, using Phusion® high-fidelity polymerase (New England Biolabs). The primers included the sequences of the RE sites to utilize of the pFastBac1 multiple cloning site. We obtained a polyhedrin and P10 knock-out pFastBac Dual vector by subjecting it to restriction digestion at the XhoI and BamHI sites, which flank the two promoters.

The obtained pFastBac1 or Dual knock-out clones were verified by Sanger sequencing and the insert was transferred into DH10Bac competent bacteria (Thermo Fisher Scientific) via transposition. The recombinant baculovirus genome i.e., bacmid, was isolated from the obtained colonies and the presence of the insert was verified by PCR.

### Animals, cell lines, transfection, VLP expression and culturing conditions

Common house cricket (*Acheta domesticus*) wingless juveniles of 10 to 15 mm in length were used, of mixed genders. 100 individuals were subjected to the initial inoculation studies to establish that AdSDV can be taken up by contaminated food. We infected 500 individuals by direct fat body injections to investigate the viral pathogenesis as well as to purify AdSDV particles from. 50 individuals were involved in direct fat body injections with either the AdSDV-Bac-*up*-ORF1 or the AdSDV-Bac-*up*-ORF3, respectively. Transfection with these plasmids were carried out by 2 mg/ml DEAE-dextran to 1 μg bacmid DNA.

Sf9 (ATCC CRL-1711) were obtained directly from the American Type Culture Collection. Suspension cultures were maintained in SF900 II medium (Gibco) in a serum-free system at 28 °C. Cellfectin II Reagent (Invitrogen) was used for DNA transfection at a cell density of $8 \times 10^5$ per well. The culturing medium was aspired and replaced by seeding medium of Grace's complete insect medium supplemented with 5% FBS (Gibco) and Graces's unsupplemented insect medium, mixed at a ratio of 1:6, respectively. After adding the transfection reagent−DNA mixture to the wells, cells were incubated for 5 h. The aspired transfection medium was replaced with SF900 II medium. Cells were checked daily for signs of CPE and the whole culture was collected when 70% of the cells detached from the dish or showed granulation. This was followed by three cycles of freeze−thaws on dry ice and 200 μL of this passage 1 (P1) stock was transferred to 25 mL of fresh Sf9 suspended cell culture in polycarbonate Erlenmeyer flasks (Corning) at the density of $2.5 \times 10^6$ cells/mL, to create the P2 stock.

### Transcription studies

Purified AdSDV-injected house crickets were collected three days post inoculation and total RNA was extracted using the Direct-zol RNA MiniPrep Kit (Zymo Research), where the denaturation step was executed by adding TRIzol Reagent (Thermo Fisher Scientific). RNA was treated by digestion with the TURBO DNA-free Kit (Ambion) to get rid of residual DNA contamination, as well as subjected to a control PCR for the remaining DNA fragments. Reverse transcription was performed only on entirely DNA-negative preparations using the SuperScript IV or the SuperScript III enzymes (Thermo Fisher Scientific), supplemented with random nonamers (Sigma-Aldrich). To avoid false detection of splicing, the isolated RNA was subjected to dephosphorylation by adding Antarctic phosphatase (New England Biolabs) and incubated for 30 min at 37 °C. Primers were designed at the following nt positions of the AdSDV NS segment: 594 (forward), 750 (reverse), 941 (reverse), 1269 (forward), 1358 (reverse), 2202 (forward and reverse), 2313 (forward), 2538 (reverse). Primers targeting mRNA of the VP segment were positioned at the following nt locations: 614 (reverse), 560 (forward), 1249 (forward), 1614 (forward and reverse), 1800 (forward and reverse), 1971 (reverse), 2611 (forward and reverse). These primers were utilized to amplify specific regions of the cDNA by PCR.

Anchored oligo(dT) primers were used with the 2202 and 2313 forward primers for the NS segment, and the 1249 and 2611 forward primers in case of the VP segment for 3' RACE (rapid amplification of cDNA ends). To perform 5' RACE to map transcription start sites, we designed adapters with the sequence of 5'−Phos-GCUGAUGGCGAUGAACACUGCGUUUGCUGGCUUUGAUGAAA−3'. RNA was subjected to dephosphorylation by alkaline calf intestinal phosphatase (New England Biolabs), followed by phenol-chloroform extraction of the dephosphorylated RNA. We utilized tobacco acid pyrophosphatase

(Ambion) to remove 5′ RNA caps. After the ligation of adapters using T4 RNA ligase 1 (New England Biolabs), reverse transcription was executed. PCR was performed with the re-adapter primers together with oligos 750 and 941 reverse in case of the NS segment and 614, 1614 and 1971 reverse for the VP segment. To find out which polyadenylation signal belongs to which promoter, the same re-adapter primer was used with 20-nt-long oligos, of which 15 nucleotides corresponded with those located directly upstream the polyA tail, also including a 5 nt-long polyT sequence. All PCRs were performed using Phusion Hot Start Flex DNA Polymerase (New England Biolabs) in a 25 µL final reaction volume, including 2 µL of purified cDNA target, 0.5 µL of both primers in 50 pmol concentration, 0.5 µL dNTP mix with 8 µmol of each nucleotide, 0.75 µL of 50 mM MgCl2 solution, and 0.25 µL of enzyme. PCR reactions were executed under a program of 5 min denaturation at 95 °C followed by 35 cycles of 30 s denaturation at 95 °C, 30 s annealing at 48 °C, and 1 or 2 min of elongation at 72 °C. The final elongation step was 8 min long at 72 °C. In case of the 5′RACE reactions, 0.5 µL of enzyme was used and the number of cycles was reduced to 25. For the 3′RACE, the reaction was supplemented with 1 µL of 50 mM MgCl2 and the annealing step was left out.

Total mRNA was also purified from Sf9 cell cultures transfected by the AdSDV-Bac-VP-ORF1, AdSDV-Bac-VP-ORF3 and AdSDV-VP-P42 bacmid constructs, respectively. Purified mRNA was reverse-transcribed and subjected to PCR amplification, so that mRNA expression of these constructs could be verified, even in the absence of VLPs.

## Protein expression and purification of VLPs and infectious virus

The AdSDV-Bac-VP-ORF1 P2 baculovirus stocks were incubated for at least five days and monitored for CPE every day. When at least 70% of the cells showed signs of CPE, the culture was collected, centrifuged at $3000 \times g$ for 15 min, and the pelleted cells disrupted by three cycles of freeze–thaws on dry ice. This lysed cell pellet was then resuspended in 1 mL of 1×TNTM pH8 (50 mM Tris pH8, 100 mM NaCl, 0.2% Triton X-100, 2 mM MgCl2) and centrifuged again. Supernatant was mixed back with the cell culture supernatant and was subjected to treatment with 250 units of Benzonase Nuclease (Sigma-Aldrich) per every 10 mL. The liquid was mixed with 1× TNET pH8 (50 mM Tris pH8, 100 mM NaCl, 0.2% Triton X-100, 1 mM EDTA) in a 1:1 ratio and concentrated on a cushion of 20% sucrose in TNET, using a type 60 Ti rotor for 3 h at 4 °C at 203,347 x g on a Beckman Coulter S class ultracentrifuge. The pellet was resuspended in 1 mL of 1×TNTM pH8 and after overnight incubation purified on a 5 to 40% sucrose step gradient for 3 h at 4 °C at 209,490 x g on the same instrument in an SW 41 Ti swinging bucket preparative ultracentrifuge rotor. The visible single band was then collected by needle puncture and a 10 mL volume syringe. For purifying infectious virus, AdSDV-inoculated crickets were mechanically homogenized in 1x PBS then subjected to the same freeze-thaw cycles and lysate clearing steps. Following Benzonase treatment the cleared lysate was subjected to the very same purification steps, detailed above. To establish the buoyant density of the AdSDV capsids, the 1xTNTM-suspended pellet of the sucrose cushion step was mixed into a 1xTNTM solution, in which CsCl was previously dissolved at a concentration of 419.5 mg/mL. The CsCl suspension was then centrifuged for 24 h in an SW 41 Ti swinging bucket preparative ultracentrifuge rotor at 10 °C at 35000 rpm. The buoyant density of the obtained fractions was established using a refractometer. The aspirated fractions were dialyzed into 1× PBS at pH 7.4 to remove the sucrose or the cesium chloride.

## Genome particle quantification

Quantification of the VP and NS segments was carried out by real-time PCR amplification (qPCR), using a Bio-Rad CFX96 instrument. A 300-bp-long target sequence was amplified of both segments (nt positions 885 to 1188 for NS and 738 to 1035 for VP). For dsDNA quantification the Bio-Rad SsoAdvanced Universal SYBR Green Supermix was used,

with an amplification program of 5-min denaturation at 95 °C followed by 45 cycles of 30 s denaturation at 95 °C, 15 s annealing at 55 °C, and 30 s of elongation at 72 °C. Results were analyzed by the CFX Maestro Software (Bio-Rad).

## Differential Scanning Fluorometry (DSF) and PLA2 assay

Capsid populations at 0.1 mg/mL concentration were dialyzed into 1x universal buffer (20 mM Hepes, 20 mM MES, 20 mM sodium acetate, 0.15 M NaCl, 3.7 mM CaCl2) at pHs 7.4, 6.0, 5.5 and 4.0. 22.5 µL capsid suspension was supplemented with 2.5 µL 1% SYPRO orange dye (Invitrogen) and subjected to DSF in a Bio-Rad CFX96 qPCR instrument. From 30 °C to 99 °C, the specimen was screened at a ramp rate of 1 °C/min in steps of 0.5 °C. Fluorescence was measured as the function of temperature, plotted as -dRFU/dT vs. temperature, which was multiplied by −1 and normalized to the highest RFU value. Each run was performed in triplicates.

For the PLA2 assay, the same capsid concentration was used in 1x universal buffer, using VLPs of parvovirus B19 as the positive control, a generous gift from Renuk Lakshmanan (University of Florida). As heating the AdDV capsid to 65 °C drastically increases PLA2 activity[21], each AdSDV capsid population was subjected to this treatment for 10 min. We performed the assay using Cayman's PLA2 Colorimetric Assay Kit. The assay was run in triplicates at 28 °C for 1 h and the absorbance was measured at 414 nm.

## Protein identification by nano-LC/MS/MS

Excised gel bands were digested with sequencing grade trypsin (Promega) and were dehydrated with 1:1 v/v acetonitrile: 50 mM ammonium bicarbonate, followed by rehydration with dithiothreitol (DTT) solution (25 mM in 100 mM ammonium bicarbonate) and the addition of 55 mM iodoacetamide in 100 mM ammonium bicarbonate solution. The protease was driven into the gel pieces by rehydrating them in 12 ng/mL trypsin in 0.01% ProteaseMAX Surfactant for 5 min. The bands were then overlaid with 40 µL of 0.01% ProteaseMAX surfactant:50 mM ammonium bicarbonate and gently mixed on a shaker for 1 h. The digestion was stopped with addition of 0.5% TFA.

Nano-LC/MS/MS was performed on a Thermo Scientific Q Exactive HF Orbitrap mass spectrometer equipped with EASY Spray nanospray source (Thermo Scientific) operated in positive ion mode. The LC system was an UltiMate™ 3000 RSLCnano system from Thermo Scientific. The mobile phase A was water containing 0.1% formic acid and the mobile phase B was acetonitrile with 0.1 % formic acid. The mobile phase A for the loading pump was water containing 0.1 % trifluoracetic acid. Five microliters of the sample was injected on to a PharmaFluidics µPAC C18 trapping column at 10 µL/mL flow rate. This was held for 3 min and washed with 1% B to desalt and concentrate the peptides. PharmaFluidics 50 cm µPAC was used for chromatographic separations with the column temperature at 40 °C. A flow rate of 750 nl/min was used for the first 15 min and then the flow was reduced to 300 nl/min. Peptides were eluted directly off the column into the Q Exactive system using a gradient of 1 to 20% B over 100 min and then to 45% B in 20 min for a total run time of 150 min. The scan sequence of the mass spectrometer was based on the original TopTen™ method; the analysis was programmed for a full scan recorded between 375–1575 Da at 60,000 resolution, and an MS/MS scan at resolution 15,000 to generate product ion spectra to determine amino acid sequence in consecutive instrument scans of the fifteen most abundant peaks in the spectrum. Singly charged ions were excluded from MS2. A siloxane background peak at 445.12003 was used as the internal lock mass. All MS/MS samples were analyzed using Proteome Discoverer 2.4 (Thermo Fisher Scientific).

## In silico analyses

The complete sequence of both AdSDV segments was assembled using Staden package v2.0.0 and v4.11.2[51]. The assembled genome was

annotated, as well as the transcripts assembled and investigated in Artemis Genome Browser by the Sanger Institute[52]. To investigate conserved domains with known homologs in the derived aa sequences the SMART web application was used[53]. Structural similarity of the resolved capsid structures with those available in the RCSB Protein Data Bank (PDB) was investigated using the DALI server[54].

For phylogeny inference, alignments, incorporating the outputs of pairwise, multiple, and structural aligners, were constructed using the Expresso algorithm of T-Coffee, ran in PDB mode[55]. The constructed alignment was edited using Unipro Ugene[56]. Model selection was executed by ProtTest v2.4, suggesting the LG + I + G + F substitution model based on both the Bayesian and Akaike information criteria[57]. Bayesian inference was executed by the BEAST v1.10.4 package, using a log-normal relaxed clock with a Yule speciation prior, throughout 50,000,000 generations[58]. Convergence diagnostics were carried out using Tracer v1.7.1 of the same package, which indicated the Markov-chain Monte Carlo runs to have converged. Phylograms were edited and displayed in the FigTree 1.4.1 program of the Beast package.

## Structural studies

Three-microliter aliquots of all AdSDV capsid populations (~1 mg/mL) were applied to glow-discharged Quantifoil holey carbon grids with a thin layer of carbon (Quantifoil Micro Tools GmbH) and vitrified using a Vitrobot Mark IV (FEI) at 95% humidity and 4 °C. The quality and suitability of the grids for cryo-EM data collection were determined by screening with a 16-megapixel charge-coupled device camera (Gatan) in a Tecnai G2 F20-TWIN transmission electron microscope operated at 200 kV in low-dose mode (~20 e−/Å2) prior to data collection. For collecting the low-resolution EC1 and 2 data sets, the same microscope was used at 50 frames per 10 s with a K2 Direct Electron Detector (DED) at the University of Florida Interdisciplinary Center for Biotechnology Research electron microscopy core (RRID:SCR_019146).

High-resolution data collection was carried out at two locations: the Florida State University (FSU) for the ORF1-VLPs, EC1 and EC2 populations, and the University of California, Los Angeles (UCLA) for the EC and FC capsids. In both cases a Titan Krios electron microscope (FEI) was used, operating at 300 kV, equipped with a Gatan K3 DED at FSU and Gatan K2 DED at UCLA. At UCLA, the scope also contained a Gatan postcolumn imaging filter and a free-path slit width of 20 eV. Movie frames were recorded using the Leginon v2.0 and v3.2 semiautomated applications at both sites[59]. At FSU, the frame rate was 50 per 10 s with ~60 e−/Å2 electron dosage. At UCLA, images were collected at 50 frames per 10 s with a ~75 e−/Å2 electron dosage. All movie frames were aligned using the MotionCor2 application with dose weighting[60].

Single-particle image reconstruction was carried out by cisTEM v1.0.[61]. Micrograph quality was assessed by CTF estimation using a box size of 512. A subset of micrographs with the best CTF fit values was included in further processing. Boxing particles was performed by the particle selection subroutine, at a threshold value of 2.0 to 4.0. Boxed particles were curated by 2D classification, imposing icosahedral symmetry at 50 classes. Particle classes, which failed to display a clear 2D-average were eliminated from the reconstruction. Ab initio model generation was carried out in 40 iterations under icosahedral symmetry constraints. The obtained startup volume was subjected to automatic refinement with imposed icosahedral symmetry and underwent iterations until reaching a stable resolution. The final maps were achieved by sharpening at a post cutoff B-factor of 10 to 20. The resolution of the obtained maps was calculated based on a Fourier shell correlation (FSC) of 0.143. Each map was resized to the voxel size determined in Chimera (by maximizing correlation coefficient) using the e2proc3D.py subroutine in EMAN2 and then converted to the CCP4 format using the program MAPMAN v3.6[62]. The atomic model of the ORF1-VLPs map was built directly into the density, without an initial

docked model, using Coot[63]. This atomic model was used to build the density of the other four reconstructions. Lastly, each model was refined against the map utilizing the rigid body, real space, and B-factor refinement subroutines in Phenix[64]. Visualization of the obtained maps and models was carried out by UCSF Chimera v1.13.1 and PYMOL v2.3.4.

## Reporting summary

Further information on research design is available in the Nature Portfolio Reporting Summary linked to this article.

## Data availability

Nucleic acid sequences, generated in this study, were deposited to the NCBI GenBank under accession numbers OP436269 and OP436270, for the ns- and vp segments, respectively. Protein sequences derived from this study were submitted to the NCBI non-redundant protein database under accession numbers WDB01642, WDB01643, WDB01644, WDB01645 and WDB01646. Density maps generated via cryo-electron microscopy and 3D image reconstruction were deposited to the Electron Microscopy Data bank, under identification numbers EMD-28604, EMD-28550, EMD-28553, EMD-28605 and EMD-28607. Atomic models built into these density maps are available at the RCSB Protein Data Bank, under identification numbers of 8EU5, 8ER8, 8ERK, 8EU6, 8EU7. The study also used the following publicly available atomic models of the RCSB Protein Data Bank with ID numbers 1DNV, 3N7X, 4MGU, 3P0S, 6WH3, 2CAS and 1LP3. Source data are provided with this paper. The NS1 protein sequence of the type species of all currently classified species of the Parvoviridae family were used for phylogenetic calculations, derived from the NCBI non-redundant protein database. Source data are provided with this paper.

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

## Acknowledgements

The authors would like to thank the late Dr. Mavis Agbandje-McKenna for her pioneering studies of parvovirus capsid structures. The study was funded by the Natural Sciences and Engineering Research Council of Canada Discovery grant (to P.T.) and an NIH grant R01 NIH GM082946 (to R.M.). The authors thank Micheline Letarte at INRS AFSB microscopy and the UF-ICBR Electron microscopy core for access to electron microscopes utilized for negative stain electron microscopy and cryo-electron micrograph screening. The Spirit and TF20 cryo-electron microscopes were provided by the UF College Of Medicine (COM) and Division of Sponsored Programs (DSP). We thank Dr. Hong Zhou (University of California Los Angeles) and the NIH "West/Midwest Consortium for High-Resolution Cryo Electron Microscopy" project for access to the Electron Imaging Center for Nanomachines's Titan Krios and K2 DED utilized for high-resolution data collection. Data collection at Florida State University was made possible by NIH grants S10 RR025080-01 (PI Taylor), and U24 GM116788 The Southeastern Consortium for Microscopy of MacroMolecular Machines (PI Taylor). The authors also thank Kari Basso at the University of Florida Mass Spectrometry Research and Education Center for the mass spectrometry work, funded from NIH S10 OD021758-01A1. We thank for the insect rearing help provided by Martin Holm.

## Author contributions

Conceived and designed by: P.T., J.J.P., and R.M. Data was collected by: J.J.P., H.T.P., P.C., and E.W.S. Analysis was performed by: J.J.P., H.T.P., P.T., R.M. Manuscript was written by: J.J.P., R.M., P.T. All authors declare to have complied with Nature portfolio's authorship Inclusion and Ethics policy.

## Competing interests

The authors declare no competing interest.
