## [Peer Review File · Nature Communications]

REVIEWER COMMENTS

Reviewer #1 (Remarks to the Author):

A report of a parvovirus variant found in an insect (crickets) that is quite distinct in a fundamental property from the other members of the Parvoviridae. That variation consists of the virus genome being divided into two shorter ssDNA segments, each of which apparently replicates separately through a parvovirus-like rolling hairpin mechanism that uses the host DNA polymerases, and each is packaged separately, although into the same capsid structure.

Although this is not well described in this study, this new virus is quite distinct from another ssDNA multipartite genomed virus in a different family (the Bidnavirus) which has a similar sized capsid but which packages two alternative full sized genomes which include a DNA polymerase - a little more information about the differences from that virus should be included.

Here the characterization of the new virus variant form includes sequences, transcription mapping using RNA seq, protein analysis using mass spec, and determining the structures of each capsid form (4 in total) which were separated by sucrose and CsCl gradients. Some of the viral genetic and structural features are novel, and indicate adaptations to the new form, and perhaps the larger genome that is allowed by this process. Overall the study is well conducted and the nature of the bipartite genome virus and the capsids seems to be well documented.

Overall the study is technically well done, so my comments are mainly for clarity and about presentation of technical issues. The main issue with the study is that it is very difficult to understand many of the details, and an expert virologist who is not a parvovirus specialist (and more specifically, a densovirus specialist and someone familiar with insect virus infections) is not going to be able to comprehend much of the study or the results. A thoughtful rewrite that focuses on the novel aspect of the bipartite genome and the proof of that discovery, and the unique elements that are required for the replication, production and transmission of the two encapsidated genome forms.

As much as possible the parvo/denso jargon should be translated and not abbreviated, so the finer details of the viruses features that are simply variants of those previously seen for other viruses - but not specifically associated with the bipartite genome issue - might be abbreviated or summarized, and the details perhaps saved for another report where those can be appreciated by the specialists.

Some specific issues to consider:

- 1) Line 28 - the term VP is confusing if the PLA2 is not part of the capsid - how does the PLA function in this case?

- 2) Line 30-31 - diverged in what way? 99% of the readers will know nothing about the Brevihamaparvoviruses and their transcription...
- 3) Line 31 - is this correct? It seems there is a single capsid form that (apparently randomly) packages one or other of the two genome forms - what is the third capsid population?
- 4) Line 33 - different in what way? If this means packaging through the threefold pore, that should be specified; but is that actually proven?
- 5) Paragraph starting line 61 - can this be simplified or summarized to outline the main issue (that related capsid structures are available for comparison, and describe any major differences)...the names and resolutions of so many different viruses that are likely not familiar to the reader are not useful information at this point in the manuscript.
- 6) Line 87 - what is meant by "is the first, hitherto, PV...."?
- 7) Line 92 - what is meant by this sentence - how does packaging involve the threefold and twofold axes (and is it proven that the fivefold axis is not involved?)
- 8) Line 143+ - the RNAs purified were sequenced via RNAseq? The descriptions of the RNAs and products is hard to follow, and the diagrams help a little; some rewriting here may help to clarify.
- 9) Line 184 - the high buoyancy and low buoyancy designations are not conventional parvovirus terminology, and are difficult to interpret even when thinking about their positions in the sucrose gradient. Since those are soon revealed to be empty and full capsids, it may be better to call those empty and full capsids so it is obvious what is being referred to.
- 10) Line 208 - 10 to 16 to 10 to 17 is a lot of genome and virus - are those numbers correct?
- 11) Discussion - this is rather long and densely written, and could be streamlined to focus on each of the main new discoveries being reported here, with the main focus on the splitting of the genome into two parts, the proof that is the case, the fact that this was derived from a likely conventional parvovirus, and the changes in the virus that were needed to allow that to be successful. Many of the comparisons with other related viruses is distracting, as is the constant focus on the PLA2, which seems like a minor side component of this virus's function. Much of the detailed description of the capsid morphology and architecture seems similar to many previous insect viruses, and unrelated to the bipartite form - or at least the connection is not obvious. The connection to threefold packaging is intriguing, but there seems to be no experimental evidence for that..
- 12) Line 307 - the statement is that genome segmentation is scarce, and then the many multipartite DNA viruses are mentioned - this is confusing.
- 13) Line 361 - the autonomous viruses have up to 11 (or more?) nucleotides ordered?
- 14) Line 363 - what is meant by two separate regions? is this the two genome segments or some other genome regions?
- 15) Line 367 - the stabilization at lower pH (and lack of PLA2) does not seem to fit with a similar infection pathway to the autonomous viruses..or something else is going on.

16) Line 369 - this seems to conflate the roles of the VP2 and VP1 in the regular parvoviruses; those have different routes of exposure and roles, and it is not clear which is being referred to in this case, but it seems more like the VP2 exposure in newly made full capsids, which can be cleaved to VP3, rather than the exposure of VP1 which appears to happen mainly during endocytosis.

17) Line 383 - what is this about - would the ions not be present in ordered positions?

Line 393 - what evidence is there that the threefold is used for DNA packaging..this seems like a dramatic difference for this virus for which there is only suggestive evidence.

18) Fig 6E - the DSF profiles are not shown - the data is presumably the melting (capsid disintegration?)temperature for each form?

Reviewer #2 (Remarks to the Author):

In this work, Penzes and colleagues described the isolation and comprehensive characterization of a novel parvovirus, *Acheta domesticus* segmented densovirus (AdSDV). AdSDV has a segmented, linear ssDNA genome and is unique among described parvoviruses. This virus represents a curious case of relatively recent genome segmentation and might be a good model for understanding this process more generally. The authors present a well-rounded, nicely written story, including the complex transcription profile and high-resolution structures for several different empty and genome-containing virus particle types. Comparison of different structures provided new insights into the assembly and potentially packaging mechanism in AdSDV and parvoviruses, in general.

L28: "...on two separate segmented genomes". This is incorrect. AdSDV has one genome which is partitioned on two segments. Similarly, multiple chromosomes found within human cells constitute one genome, not multiple.

L29: From the way it is written, it seems that PLA2 gene is absent from the capsid. Is this the case or did the authors mean that the PLA2 protein is not encapsidated. This might become obvious after reading the entire manuscript, but the Abstract should stand on its own.

L31: Please explain in a few words what is meant by 3 capsid populations. Structurally different? This is a bit confusing given that there are only two segments.

L53: SF3 helicase is not a sequence motif, but a domain with several motifs, at least Walker A and Walker B.

Fig. 1C: There is an orphan “B” letter between the two depicted genomic termini. Delete?

Fig. 1D: This panel is confusing. First, what is ORF3, which is not mentioned in the text? Second, what are the two small not-numbered ORFs labeled with asterisks (not explained in the figure legend)? Are they proper ORFs (with start and stop)? If not, chose some other way to depict them. Also, Consider unifying the ORF labeling – why have NS1 and NS2, but VP ORF1 and VP ORF3? Why not VP1 and VP3 (or VP2)?

L127: The second segment is shown in Fig. 1E, not 1D.

L130: The similarity between the product of ORF2 and SF1 helicases is dubious. Clearly, the ORF is too short to encode any functional helicase. I would suggest leaving out the mention of this spurious match.

L133: It would be useful to provide some additional information about Scindoambidensovirus VP1u. Is it typical for parvoviruses that PLA2 domain is in the middle of the VP ORF rather than at the terminus?

L142: Please mention briefly what was done with the total RNA once it was isolated (e.g., “... isolated and sequenced”)?

L150: Can a stretch of coding nucleotide sequence be called ORF if it does not have a start codon?

Fig. 2: Typo in AdSDV (“doemstica”). Besides, on line 82, you write “*Acheta domesticus*”, whereas in the figure it is “*domestica*”.

L183: What is meant by “showing advanced signs”?

L191-L205: Please specify whether the particles discussed correspond to purified native virions, rather than VLPs discussed in the previous paragraph. This is obvious from Fig. 3, but would be good to clarify in the text as well.

L199: Is it not surprising that VP1 gene produced a single transcript but is translated as a range of different variants, whereas VP3 gene produces a range of transcripts but none of the proteins are found in the virions?

L215: Extra “single” in “single-single particle”.

L270: Was Fig. 6C meant here?

L320: Does VP-ORF3 have a potential to fold into a decent-looking capsid protein? The authors could build and evaluate an AlphaFold model of VP-ORF3.

L374: “evolutionarily”.

The authors suggest that VP-ORF3 might provide benefits in other hosts. AdSDV does not infect the other offered cricket species, but have they considered trying some mosquitos or their cell lines (I am not asking to do these experiments for this manuscript)?

Can the authors briefly discuss how AdSDV is expected to exit from the endosomes without the PLA2 domain and perhaps draw some parallels with other PLA2-lacking parvoviruses?

SF3 domain clearly places AdSDV into the Brevihamaparvovirus genus. However, in its genome organization and gene content, AdSDV differs dramatically from other members of this genus. Are the authors comfortable with the placement of AdSDV into Brevihamaparvovirus or would they consider an exception to the established ICTV demarcation criteria? The authors could add a brief discussion on the classification.

Reviewer #3 (Remarks to the Author):

In this manuscript, the authors describe the discovery of a new virus, *Acheta domesticus* segmented densovirus (AdSDV), with an uncharacteristic bipartite DNA genome that lacks the inverted repeats (ITR) as belonging to the Parvoviridae family. Typically, the members in this family are small T=1 icosahedral viruses with a linear monopartite DNA genome with ITRs. By analyzing the genome of the virus particles isolated from common house crickets at an insect-rearing facility, authors show that AdSDV genome is

linear with a blunt end and consists of two segments. Phylogenetic and sequence motif analysis of the structural and non-structural proteins encoded by the bipartite genome show that these proteins have all the classical features characteristic of the parvovirus/densovirus proteins justifying AdSDV as a bonafide member of Parvoviridae. Despite belonging to this family, however, because of the segmented nature of the genome, the transcription strategy employed by AdSDV is different from that of the typical parvovirus. From the biochemical analysis of the particles obtained from AdSDV-infected house crickets, which form two distinct bands in the density gradient (low-buoyancy and high buoyancy), and the VLPs, produced using a baculovirus-expression system, authors show that each segment is packaged into a separate particle utilizing a strategy distinct from that proposed for other parvovirus/densovirus. High-resolution cryo-EM analysis of these particles shows that the particles in each group display the expected T=1 icosahedral assembly and the overall folding of the C-terminal core of the capsid protein. These particles also show noticeable differences in their genomic content and the subunit interactions differentially modulated by the nucleotides and the N-terminal regions of the capsid proteins supporting the authors' contention that the genome packaging strategy in AdSDV is different from that suggested for other parvoviruses/densoviruses.

Overall, this is a well-written manuscript describing the discovery and characterization of a new virus with a bipartite genome belonging to the Parvoviridae family. The genome characterization, genomic organization, transcription strategy, and analysis of the protein composition and motifs are all quite convincing in supporting the discovery. Cryo-EM analysis of the four different sets of particles is well performed. The structural analysis and interpretations convincingly demonstrate the similarities and differences between AdSDV and other members in the Parvoviridae in terms of capsid protein conformations, bound genome segments, and N-terminal tail densities and support the notion that AdSDV has different genome packaging strategy.

I have some comments that the authors' should consider:

1) The main text is a bit jargonish, replete with parvovirus terminology, which could be minimized to enhance the readability for the audience not conversant with parvovirus biology. From that point of view, one could argue that this paper is more suitable for a specialty journal.

2) Figures 4-6 have too many panels and are too small to appreciate their description in the main text. These figures, particularly those describing structural features, can be simplified and made larger and less cramped by shifting some panels to supplementary materials. Ex. the panel that shows peptide fitting the density in Fig. 4, and panel E in Fig. 5, etc.

3) Line 256-257 "... covering a ring of three histidines (His232)..." I don't see it.

4) Last sentence in the abstract, “this study provides a new perspective on ssDNA genome segmentation and on the plasticity of parvovirus biology” is a run-on sentence that does not say much. – consider revising

5) Line 38: words “protostome and deuterostome” – necessary?

Response to Reviewers

We would like to thank all the reviewers for taking their time reviewing the manuscript and helped us to improve it by providing constructive comments. We have revised the manuscript accordingly. Please find our responses below, addressing each of the queries.

Best regards,

Judit Penzes
Corresponding Author

Reviewer #1 (Remarks to the Author):

A report of a parvovirus variant found in an insect (crickets) that is quite distinct in a fundamental property from the other members of the Parvoviridae. That variation consists of the virus genome being divided into two shorter ssDNA segments, each of which apparently replicates separately through a parvovirus-like rolling hairpin mechanism that uses the host DNA polymerases, and each is packaged separately, although into the same capsid structure.

Answer: AdSDV replicates through a rolling-hairpin mechanism that uses the host DNA polymerases and the identical terminal hairpins of the two segments for packaging.

Although this is not well described in this study, this new virus is quite distinct from another ssDNA multipartite genomed virus in a different family (the Bidnavirus) which has a similar sized capsid but which packages two alternative full sized genomes which include a DNA polymerase - a little more information about the differences from that virus should be included.

Answer: Bidnaviruses belong to another virus family derived from Polinton/Maverick lineage (Ref 58 in line 318). Bidnavirus structural proteins, in concordance with the significantly bigger size of the two genome segments (6 to 6.5 kb), are very large (about 130 kDa) and express minor structural proteins that seem to be derived from dsRNA reoviruses and dsDNA baculoviruses. In contrast to parvoviruses, bidnaviruses code for their own DNA polymerase and use a terminal protein-initiated replication, as opposed to the rolling hairpin mechanism of parvoviruses, including AdSDV. We added a couple of lines on this to the text, see lines 318-320.

Here the characterization of the new virus variant form includes sequences, transcription mapping using RNA seq, protein analysis using mass spec, and determining the structures of each capsid form (4 in total) which were separated by sucrose and CsCl gradients. Some of the viral genetic and structural features are novel, and indicate adaptations to the new form, and perhaps the larger genome that is allowed by this process. Overall the study is well conducted and the nature of the bipartite genome virus and the capsids seems to be well documented.

Overall the study is technically well done, so my comments are mainly for clarity and about presentation of technical issues. The main issue with the study is that it is very difficult to understand many of the details, and an expert virologist who is not a parvovirus specialist (and more specifically, a densovirus specialist and someone familiar with insect virus infections) is not going to be able to comprehend much of the study or the results. A thoughtful rewrite that focuses on the novel aspect of the bipartite genome and the proof of that discovery, and the unique elements that are required for the replication, production and transmission of the two encapsidated genome forms.

As much as possible the parvo/denso jargon should be translated and not abbreviated, so the finer details of the viruses features that are simply variants of those previously seen for other viruses - but not specifically associated with the bipartite genome issue - might be abbreviated or summarized, and the details perhaps saved for another report where those can be appreciated by the specialists.

Answer: the clarifications requested will make it easier to understand by the general reader. We appreciate the request of omitting details concerning the family exclusively, although we must point out that AdSDV is a rather unique member of this family in addition to its bipartite replication strategy. These features encompass the capsid architecture itself; hence we would like to keep the structural description of the capsid. We tried, however, to shorten and simplify these paragraphs.

In addition:

Line 27-28 change to: cassettes on the two separate segments of the bipartite genome.

We have changed this as requested.

Some specific issues to consider:

1) Line 28 - the term VP is confusing if the PLA2 is not part of the capsid - how does the PLA function in this case?

Answer: An excellent point raised by the reviewer. We still use the term "VP" as this ORF is also located on the same segment as the major capsid protein encoding ORF, in which case VP-ORF3 is referring to the third ORF of the VP segment. In an additional experiment, we expressed the nonstructural (ns) PLA2 in a baculovirus vector and observed, compared to transfection with the control vector containing only VP ORF1, a mass die-off of the transfected crickets as well as transfected Sf9 cells in a few days, possibly due to toxicity or programmed cell death. This is a common strategy in virus spread (e.g. Danthi (2016): Viruses and the diversity of cell death. Annual Rev Virol 3:533 <https://doi.org/10.1146/annurev-virology-110615-042435>). This raises the question whether the parvoviruses with a sPLA2 in the capsid also have a dual function and the impact these motifs may have in AAV gene therapy. This issue is especially intriguing in case of densoviruses, which encode the VP1u as a separate exon. Theoretically, these *cap1* ORFs could also be expressed individually in unspliced transcripts.

This experiment was added to “Results” under lines 190-195 and the observation has been integrated in the “Discussion” under lines 330-335.

2) Line 30-31 - diverged in what way? 99% of the readers will know nothing about the Brevihamaparvoviruses and their transcription...

Answer: In complexity of transcription strategy. Bayesian phylogeny inference (shown in Figure 2) confirms the genus affiliation of AdSDV. We feel it would be superfluous to go into details of the transcription of the Brevihamaparvoviruses (not a review). We have, nonetheless, rephrased this sentence for clarity and comprehensibility for a broader audience.

3) Line 31 - is this correct? It seems there is a single capsid form that (apparently randomly) packages one or other of the two genome forms - what is the third capsid population?

Answer: Capsid populations:

1. Full capsids, packaging the NS or VP segment and, in a separate capsid their respective complement (upon extraction they anneal to a double-stranded DNA)
2. Empty capsids with lower buoyancy in CsCl, displaying a tight fivefold channel
3. Empty capsids with somewhat higher buoyancy in CsCl, displaying a completely open fivefold channel

Line 33 - different in what way? If this means packaging though the threefold pore, that should be specified; but is that actually proven?

Answer: change “reveal a genome packaging mechanism, which differs from other parvoviruses.” to: “reveal a novel genome packaging mechanism among parvoviruses.” See line 34.

Paragraph starting line 61 - can this be simplified or summarized to outline the main issue (that related capsid structures are available for comparison, and describe any major differences)...the names and resolutions of so many different viruses that are likely not familiar to the reader are not useful information at this point in the manuscript.

Answer: As opposed to the many structural studies carried out on vertebrate parvoviruses, these are the only densoviral capsid structures and they are often compared with that of AdSDV throughout the manuscript. As only so few are available for such a populous parvovirus group and all have rather distinct features, therefore we feel we cannot omit them.

Line 87 - what is meant by "is the first, hitherto, PV...."?

Answer: change to: Up to now, AdSDV is the first PV to harbor a bipartite genome

Line 92 - what is meant by this sentence - how is does packaging involve the threefold and twofold axes (and is it proven that the fivefold axis is not involved?)

Answer: This is the last paragraph of the Introduction stating what is coming in the Results, not the data, evidence and prove. Although there is no direct experimental evidence that this is the case, there is indirect evidence in the structures that point to this direction, see Discussion lines 353-362

Line 143+ - the RNAs purified were sequenced via RNAseq? The descriptions of the RNAs and products is hard to follow, and the diagrams help a little; some rewriting here may help to clarify.

Answer: NGS-based RNAseq was not performed. RNA was isolated and purified from insects and cells as described in lines 457-470. After control experiments with PCR that it was DNA-free, reverse transcription, avoiding false splicing, and DNA sequencing was performed. The combination of Figure 2 and Lines 449-486 should facilitate the understanding of the experiment and results.

Line 184 - the high buoyancy and low buoyancy designations are not conventional parvovirus terminology, and are difficult to interpret even when thinking about their positions in the sucrose gradient. Since those are soon revealed to be empty and full capsids, it may be better to call those empty and full capsids so it is obvious what is being referred to.

Answer: The reviewer is correct and corrections has been made throughout the manuscript, now calling them empty capsid (EC) and full capsid (FC) fractions.

Line 208 - 10 to 16 to 10 to 17 is a lot of genome and virus - are those number correct?

Answer: At 260 nm and 1 cm pathway an optical density of 7 corresponds to 10^{14} full particles. Although the qPCR experiments were performed in triplicates and we did not observe any alarming standard deviation values, the reliability of quantification over 10^{14} particles is compromised by the detection limit of the instrument. What we know for sure is that we had more particles than this number. We have revised this figure and text to indicate the unreliability of the methodology in this concentration range, see lines 215-220.

Discussion - this is rather long and densely written, and could be streamlined to focus on each of the main new discoveries being reported here, with the main focus

on the splitting of the genome into two parts, the proof that is the case, the fact that this was derived from a likely conventional parvovirus, and the changes in the virus that were needed to allow that to be successful. Many of the comparisons with other related viruses is distracting, as is the constant focus on the PLA2, which seems like a minor side component of this viruses function. Much of the detailed description of the capsid morphology and architecture seems similar to many previous insect viruses, and unrelated to the bipartite form - or at least the connection is not obvious. The connection to threefold packaging is intriguing, but there seems to be no experimental evidence for that..

We edited the discussion and tried to streamline it, albeit we must point out that AdSDV is not only a unique parvovirus because of its bipartite replication strategy, but provides the first structural insight into the Brevihamaparvovirus genus, as well as only hitherto the second structural characterization of a member of the little-understood Hamaparvovirinae subfamily. The description of its multimer interactions, apart from demonstrating and unforeseen diversity in parvovirus capsid architecture, are required in order to introduce and understand the genome packaging model of this virus, that is fundamentally different from the canonical *Parvo-* and *Densovirinae* one.

Line 307 - the statement is that genome segmentation is scarce, and then the many multipartite DNA viruses are mentioned - this is confusing.

Answer: We have rephrased the sentence, see lines 307-308.

Line 361 - the autonomous viruses have up to 11 (or more?) nucleotides ordered?

Answer: the reviewer's query is not clear. The C-terminus of the local subunits comprising the threefold axes separated by the twofold symmetry axes stretches far and bends underneath the twofold symmetry axis, which is locked by ordered DNA (six ordered nucleic acid bases ($\sigma=3$)) in the genome-filled particles. In autonomous parvoviruses of the genus Protoparvovirus display large number of icosahedrally ordered nucleotide stacks, which are possible to model, see for reference:

- Llamas-Saiz AL, Agbandje-McKenna M, Wikoff WR, Bratton J, Tattersall P, Rossmann MG. Structure determination of minute virus of mice. *Acta Crystallogr D Biol Crystallogr*. 1997 Jan 1;53(Pt 1):93-102. doi: 10.1107/S09074444996010566. PMID: 15299974.
- Tsao J, Chapman MS, Agbandje M, Keller W, Smith K, Wu H, Luo M, Smith TJ, Rossmann MG, Compans RW, et al. The three-dimensional structure of canine parvovirus and its functional implications. *Science*. 1991 Mar 22;251(5000):1456-64. doi: 10.1126/science.2006420. PMID: 2006420.

There is no current information available on the genome structure of autonomous parvoviruses of other Parvovirinae genera.

Line 363 - what is meant by two separate regions? is this the two genome segments or some other genome regions?

Answer: two ssDNA regions of the same segment. Based on the ordered density, it is clear that these two ordered nucleotide strings are not directly linked by the sugar-phosphate backbone, hence they must be located on distant regions of the same segment, packaged by given particle.

Line 367 - the stabilization at lower pH (and lack of PLA2) does not seem fit with a similar infection pathway to the autonomous viruses..or something else is going on.

Answer: It fits, however, the mechanism of another parvovirus that lacks a PLA2 and uses an alternate endosomal egress mechanism, driven by a non-PLA2-like membrane penetrating peptide, which is controlled by the presence or absence of divalent cations (Penzes et al., PNAS 117: 20211 (2020)). The absence of a capsid-linked PLA2 does not necessarily exclude endosomal trafficking. Many virus families follow this trafficking route yet lack a PLA2 domain.

Line 369 - this seems to conflate the roles of the VP2 and VP1 in the regular parvoviruses; those have different routes of exposure and roles, and it is not clear which is being referred to in this case, but it seems more like the VP2 exposure in newly made full capsids, which can be cleaved to VP3, rather than the exposure of VP1 which appears to happen mainly during endocytosis.

Answer: The reviewer is correct for the members of Parvovirinae and Densovirinae. Many members of the Hamaparvovirinae, however, express only one capsid protein. Here, we did not distinguish the different VPs, as only one of them seems to be available at the time of capsid assembly, with the other size being only the product of virion maturation.

Line 383 - what is this about - would the ions not be present in ordered positions?

Answer: Most probably yes, but they could not be averaged, hence will be absent from the structure.

Line 393 - what evidence is there that the threefold is used for DNA packaging. This seems like a dramatic difference for this virus for which there is only suggestive evidence.

Answer: Not yet solid evidence, we state: “might involve the threefold annulus for genome entry, after which the genome is attached to the luminal twofold axis by the VP C-termini” and in view of the nature of the five-fold channel. The assumption derives from the structural study of GmDV, where the size of this opening exceeds the one at the fivefold axes, which is also linked to a surprisingly short fivefold channel. Moreover, the threefold annulus is always open, as opposed to the density filled GmDV fivefold channel. In case

of AdSDV the existence and similar nature of the annulus in addition to the threefold axis-related C-terminal tails, which have no involvement in creating the shell itself, are in concordance with this hypothesis.

Fig 6E - the DSF profiles are not shown - the data is presumably the melting (capsid disintegration?) temperature for each form?

The complete DSF profiles for each capsid populations under each pH circumstance are shown in supplemental figure S8. Melting temperature profiles in relation to pH are shown in Figure 10E, which were derived from the DSF data.

Reviewer #2 (Remarks to the Author):

In this work, Penzes and colleagues described the isolation and comprehensive characterization of a novel parvovirus, *Acheta domesticus* segmented densovirus (AdSDV). AdSDV is has a segmented, linear ssDNA genome and is unique among described parvoviruses. This virus represents a curious case of relatively recent genome segmentation and might be a good model for understanding this process more generally. The authors present a well-rounded, nicely written story, including the complex transcription profile and high-resolution structures for several different empty and genome-containing virus particle types. Comparison of different structures provided new insights into the assembly and potentially packaging mechanism in AdSDV and parvoviruses, in general.

L28: "...on two separate segmented genomes". This is incorrect. AdSDV has one genome which is partitioned on two segments. Similarly, multiple chromosomes found within human cells constitute one genome, not multiple.

Answer: correct: on two segments of the bipartite genome, now revised in line 29.

L29: From the way it is written, it is seems that PLA2 gene is absent from the capsid. Is this the case or did the authors mean that the PLA2 protein is not encapsidated. This might become obvious after reading the entire manuscript, but the Abstract should stand on its own.

Answer: We have revised this sentence, see line 29-31.

L31: Please explain in a few words what is meant by 3 capsid populations. Structurally different? This is a bit confusing given that there are only two segments.

Answer: Two empty capsid populations of two different buoyant densities, which also differ in fivefold channel conformation; full genome-packaging particles. Moreover, we

also resolved the structure the vpORF1-encoded virus-like particles.

L53: SF3 helicase is not a sequence motif, but a domain with several motifs, at least Walker A and Walker B.

Answer: change “encompasses a superfamily 3 (SF3) helicase domain, which is the only highly conserved protein sequence motif throughout the entire family [32, 34].” To encompass a superfamily 3 (SF3) helicase domain, which has the only highly conserved protein sequence motifs throughout the entire family [32, 34]., see lines 53-55.

Fig. 1C: There is an orphan “B” letter between the two depicted genomic termini. Delete?

Answer: It has been deleted.

Fig. 1D: This panel is confusing. First, what is ORF3, which is not mentioned in the text? Second, what are the two small not-numbered ORFs labeled with asterisks (not explained in the figure legend)? Are they proper ORFs (with start and stop)? If not, chose some other way to depict them. Also, Consider unifying the ORF labeling – why have NS1 and NS2, but VP ORF1 and VP ORF3? Why not VP1 and VP3 (or VP2)?

Answer: See Line 160: a small auxiliary ORF, ORF3, (is spliced) with the C-terminal portion of the NS1 ORF, potentially encoding the 20-kDa-sized NS1-C. (color-coded in Fig 1D).

ORF3 is a small ORF, which has no detectable homologue in the NCBI databases. It is now mentioned in the text, see line 134-135. We would like to keep the current labeling system, however, as NS1 and NS2 are clear homologues of the ORFs with the same name in other members of the genus, yet VP1, VP2 and so on would be confusing, as these usually mean multiple proteins expressed by one (or sometimes two) ORFs, differing only in N-terminal extensions.

The small ORFs marked by the asterisk indeed lack an ATG start codon and function as small exons. This has been clarified in the figure legend.

L127: The second segment is shown in Fig. 1E, not 1D.

Answer: corrected.

L130: The similarity between the product of ORF2 and SF1 helicases is dubious. Clearly, the ORF is too short to encode any functional helicase. I would suggest leaving out the mention of this spurious match.

Answer: We have omitted the description of this match from the revised manuscript.

L133: It would be useful to provide some additional information about Scindoambidensovirus VP1u. Is it typical for parvoviruses that PLA2 domain is in the middle of the VP ORF rather than at the terminus?

Answer: PVs of the *Parvovirinae* possess the PLA2 domain at the N-termini; however many densoviruses (but also chipmunk PV and B19) have the PLA2 125 to 180 amino acids downstream (see Zadori; Dev Cell 1: 291 (2001)).

Members of the *Scindoambidensovirus* genus are characterized by a split VP-encoding ORF, which gives rise to the VP1 minor capsid protein via a spliced transcript, uniting the PLA2-including cap1 with a separate ORF, cap2. Cap2 gives rise to VPs2 and 3 (if present), as well as the major capsid protein, VP4. Similar VP expression strategy occurs in four more Densovirinae genera, i.e. *Hemiambidensovirus*, *Pefuambidensovirus*, *Muscodensovirus* and *Blattambidensovirus*. The position of the PLA2 domain within the VP1 ORF varies greatly, as positioned N-terminally in *Muscodensovirus*, central in *Scindo-* and *Hemiambidensovirus* and C-terminal in *Blatt-* and *Pefuambidensovirus*. Because of the length of the current manuscript, unfortunately, we cannot go into details on the various split VP expression strategies of densoviruses.

L142: Please mention briefly what was done with the total RNA once it was isolated (e.g., “... isolated and sequenced”)?

Answer: RNA was isolated and purified from insects and cells as described in lines 449-486. After control experiments with PCR that it was DNA-free, reverse transcription, avoiding false splicing was performed. We performed PCR reactions with primers, listed in the Methods section, using the cDNA as a template. These PCR products were sequenced and analyzed. The combination of Figure 2 and Lines 457-470 should facilitate the understanding of the experiment and results.

L150: Can a stretch of coding nucleotide sequence be called ORF if it does not have a start codon?

Answer: Not on its own. However, all of these frames become part of a complete ORF through splicing. We included these frames as “exons” as well as in the figure description of the NS segment (Fig. 1D).

Fig. 2: Typo in AdSDV (“doemstica”). Besides, on line 82, you write “Acheta domesticus”, whereas in the figure it is “domestica”.

Answer: Corrected doemstica to domesticus as this is the correct zoological term.

L183: What is meant by “showing advanced signs”?

Answer: crickets that are showing clinical signs, which prevent them from performing their original behavior routine, i.e. incapable of movement and feeding. We have changed this term to lethargic, see line 196.

L191-L205: Please specify whether the particles discussed correspond to purified native virions, rather than VLPs discussed in the previous paragraph. This is obvious from Fig. 3, but would be good to clarify in the text as well.

Answer: The gradient had two distinct bands of particles from infected crickets, i.e., in the 20% sucrose fraction (low density or empty capsids (EC)) as well as in the 25% fraction (high density or full capsids (FC)) (full capsids have a higher density and a higher mass and would accumulate at a higher sucrose concentration. This is clarified at the end of this paragraph, see lines 196-198 and 215-220.

L199: Is it not surprising that VP1 gene produced a single transcript but is translated as a range of different variants, whereas VP3 gene produces a range of transcripts but none of the proteins are found in the virions?

Answer: It is common that nonstructural proteins like vpORF3 have different forms for different activities. This may further confirm the nonstructural nature of this protein, as typically these derive from spliced transcripts in parvoviruses, as opposed to the structural proteins, predominantly expressed by leaky scanning. Many hamaparvoviruses express one single VP, though.

L215: Extra “single” in “single-single particle”.

Answer: corrected

L270: Was Fig. 6C meant here?

Answer: yes, and now corrected. These figures have been re-numbered as more of these were added.

L320: Does VP-ORF3 have a potential to fold into a decent-looking capsid protein? The authors could build and evaluate an AlphaFold model of VP-ORF3.

Answer: It does not. This prediction has been done and now added as Figure 4 and lines 223-227. This new figure makes it clear that this protein lacks a jelly roll-like fold and that it is mostly helical in nature. Despite of the overall low confidence and pLDDT score, the PLA2 domain folds confidently, displaying its canonical motifs in concordance with those of functional PLA2s.

L374: “evolutionarily”.

Corrected.

The authors suggest that VP-ORF3 might provide benefits in other hosts. AdSDV does not infect the other offered cricket species, but have they considered trying some mosquitos or their cell lines (I am not asking to do these experiments for this manuscript)?

Answer: Yes, these experiments have been done, but due to the complex nature of the manuscript, this negative result was omitted. We tried to both infect and transfect Sf9 cells of lepidopteran origin, S2 cells of dipteran origin and the C6/36 mosquito cell line of similarly dipteran origin. C6/36 is permissive to dipteran brevihamaparvoviruses.

Can the authors briefly discuss how AdSDV is expected to exit from the endosomes without the PLA2 domain and perhaps draw some parallels with other PLA2-lacking parvoviruses?

Answer: A divalent cation-dependent, non-PLA2-linked membrane penetrating mechanism has been proposed for *Penaeus monodon* metallodensovirus (Penzes et al., PNAS 117: 20211 (2020)). Although PmMDV has a clear Arg-rich membrane penetrating domain in its VP N-terminus, which shows striking sequential and structural similarity to the membrane penetrating apparatus of nodaviruses, no such domain could be revealed in case of AdSDV. Our hitherto unpublished results, however, indicate that *Aedes albopictus* densovirus, a brevihamaparvovirus with no PLA2 homologue in its genome, displays the exact same pH-related melting temperature profile. The absence of a capsid-linked PLA2 does not necessarily exclude endosomal trafficking, though as several ssRNA virus families follow this trafficking route yet lack a PLA2-like domain. It is fair to assume that PLA2-mediated membrane lysis is only one strategy out of several yet uncharacterized ones within the more and more populous *Parvoviridae*. The precise mechanism of this for AdSDV is yet to be characterized. We, however, feel that speculating on a potential mechanism exceeds the scope and editorial limits of this manuscript.

SF3 domain clearly places AdSDV into the Brevihamaparvovirus genus. However, in its genome organization and gene content, AdSDV differs dramatically from other members of this genus. Are the authors comfortable with the placement of AdSDV into Brevihamaparvovirus or would they consider an exception to the established ICTV demarcation criteria? The authors could add a brief discussion on the classification.

Answer: The *Brevihamaparvovirus* ancestry of AdSDV is easily traceable and detectable, suggesting that the bipartite genome is only the consequence of acquiring an additional transcription unit, which would make packaging of this extended genome impossible. Although the existence of AdSDV may suggest that there are other bipartite parvoviruses yet undiscovered, it would not be surprising if these were in fact only distantly related to AdSDV and evolved the same strategy independently. In fact, it might even beg the question for the bipartite *Bidnaviridae*, whether the case of *Bombyx mori* bidensovirus is also an isolated one due to interfamily recombination events and the vast, yet undiscovered/unclassified majority of these polB-encoding ssDNA viruses are in fact monopartite. As the *Parvoviridae* study group considers the SF3 domain to be the major

determinant of taxonomy affiliation, we believe this has priority over a potentially convergent evolutionary trait, such as the bipartite genome. This is in concordance with the placement of AdSDV within genus *Brevihamaparvovirus*. Unfortunately, discussing this otherwise intriguing taxonomy challenge would also exceed the editorial limits of this manuscript.

Reviewer #3 (Remarks to the Author):

In this manuscript, the authors describe the discovery of a new virus, *Acheta domesticus* segmented densovirus (AdSDV), with an uncharacteristic bipartite DNA genome that lacks the inverted repeats (ITR) as belonging to the Parvoviridae family. Typically, the members in this family are small T=1 icosahedral viruses with a linear monopartite DNA genome with ITRs. By analyzing the genome of the virus particles isolated from common house crickets at an insect-rearing facility, authors show that AdSDV genome is linear with a blunt end and consists of two segments. Phylogenetic and sequence motif analysis of the structural and non-structural proteins encoded by the bipartite genome show that these proteins have all the classical features characteristic of the parvovirus/densovirus proteins justifying AdSDV as a bonafide member of Parvoviridae. Despite belonging to this family, however, because of the segmented nature of the genome, the transcription strategy employed by AdSDV is different from that of the typical parvovirus. From the biochemical analysis of the particles obtained from AdSDV-infected house crickets, which form two distinct bands in the density gradient (low-buoyancy and high buoyancy), and the VLPs, produced using a baculovirus-expression system, authors show that each segment is packaged into a separate particle utilizing a strategy distinct from that proposed for other parvovirus/densovirus. High-resolution cryo-EM analysis of these particles shows that the particles in each group display the expected T=1 icosahedral assembly and the overall folding of the C-terminal core of the capsid protein. These particles also show noticeable differences in their genomic content and the subunit interactions differentially modulated by the nucleotides and the N-terminal regions of the capsid proteins supporting the authors' contention that the genome packaging strategy in AdSDV is different from that suggested for other parvoviruses/densoviruses.

Overall, this is a well-written manuscript describing the discovery and characterization of a new virus with a bipartite genome belonging to the Parvoviridae family. The genome characterization, genomic organization, transcription strategy, and analysis of the protein composition and motifs are all quite convincing in supporting the discovery. Cryo-EM analysis of the four different sets of particles is well performed. The structural analysis and interpretations convincingly demonstrate the similarities and differences between AdSDV and other members in the Parvoviridae in terms of capsid protein conformations, bound genome segments, and N-terminal tail densities and support the notion that AdSDV has different genome packaging strategy.

I have some comments that the authors' should consider:

1) The main text is a bit jargonish, replete with parvovirus terminology, which could be minimized to enhance the readability for the audience not conversant with parvovirus biology. From that point of view, one could argue that this paper is more suitable for a specialty journal.

Answer: we have carefully edited the manuscript in order to make it accessible to a wide range of readers. We believe that even the parvovirus-specific aspects of this manuscript may be of interest for a broader audience, as this family is increasingly astonishing in its diversity and offers intriguing aspects that may offer a new perspective on the structure and evolution of other small virus families, which are under similar evolutionary constraints as the *Parvoviridae*.

2) Figures 4-6 have too many panels and are too small to appreciate their description in the main text. These figures, particularly those describing structural features, can be simplified and made larger and less cramped by shifting some panels to supplementary materials. Ex. the panel that shows peptide fitting the density in Fig. 4, and panel E in Fig. 5, etc.

Answer: We have divided these figures up for better visibility. Figure 4 is now figure 5 and 6. Unfortunately, the rest of these figures are direct comparisons hence dividing them would make direct comparison difficult.

3) Line 256-257 “.... covering a ring of three histidines (His232)...” I don't see it.

Answer: We are now showing this capsid feature from the luminal view, which makes the visibility of these easier. See updated Figure 9.

4) Last sentence in the abstract, “this study provides a new perspective on ssDNA genome segmentation and on the plasticity of parvovirus biology” is a run-on sentence that does not say much. – consider revising

Answer: This is now revised, see lines 34-36.

5) Line 38: words “protostome and deuterostome” – necessary?

Answer: It is now deleted.

REVIEWERS' COMMENTS

Reviewer #1 (Remarks to the Author):

I read through the revision, and it seems much clearer and still significant, and I have no further comments.

Reviewer #2 (Remarks to the Author):

I thank the authors for addressing all of my comments. I have no further questions.